# Linearly Mapping from Image to Text Space

**Jack Merullo, Louis Castricato, Carsten Eickhoff, Ellie Pavlick**
Department of Computer Science
Brown University
Providence, RI, USA
{jack_merullo,louis_castricato,carsten,ellie_pavlick}@brown.edu

## Abstract

The extent to which text-only language models (LMs) learn to represent features of the non-linguistic world is an open question. Prior work has shown that pre-trained LMs can be taught to caption images when a vision model's parameters are optimized to encode images in the language space. We test a stronger hypothesis: that the conceptual representations learned by frozen text-only models and vision-only models are similar enough that this can be achieved with a linear map. We show that the image representations from vision models can be transferred as continuous prompts to frozen LMs by training only a single linear projection. Using these to prompt the LM achieves competitive performance on captioning and visual question answering tasks compared to models that tune both the image encoder and text decoder (such as the MAGMA model). We compare three image encoders with increasing amounts of linguistic supervision seen during pretraining: BEIT (no linguistic information), NF-ResNET (lexical category information), and CLIP (full natural language descriptions). We find that all three encoders perform equally well at transferring visual property information to the language model (e.g., whether an animal is large or small), but that image encoders pretrained with linguistic supervision more saliently encode category information (e.g., distinguishing hippo vs. elephant) and thus perform significantly better on benchmark language-and-vision tasks. Our results indicate that LMs encode conceptual information structurally similarly to vision-based models, even those that are solely trained on images. Code is available here: https://github.com/jmerullo/limber

## 1 Introduction

Much recent work in NLP has revolved around studying the limits on representational capacity incurred by training on form-only text data, as discussed in Bender & Koller (2020). Tied to this argument is the idea that without explicit *grounding*, language models are not inclined to learn conceptual representations of language that reflect the rich conceptual knowledge that humans gain from interacting with the physical, non-linguistic world. Despite this, there have been remarkable findings in large language models' abilities to generalize to and reason about non-linguistic phenomena (Tsimpoukelli et al., 2021; Eichenberg et al., 2021; Li et al., 2021; Patel & Pavlick, 2022). Thus, an open question in the field is to what extent (if at all) a language model trained on text-only data can learn aspects of the physical world. In this paper, we test a specific hypothesis about the relationship

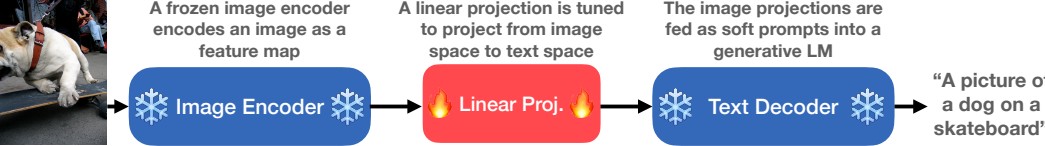

Figure 1: We train linear projections from image representations into the input space of a language model to produce captions describing images. We find that LMs can describe the contents of most image representations, but performance varies based on the type of image encoder used.

between language model and image encoder representations: that these conceptual representations can be approximately mapped to one another through a linear transformation. To do this, we train a single linear layer to project from the representation space of images into the language space of a generative LM without tuning any other model parameters, which we call `LiMBeR`: Linearly Mapping Between Representation spaces. That is, we linearly transform an image representation into "soft prompts"–vector(s) in the embedding space that do not correspond to discrete language tokens (Lester et al., 2021). The weights of this linear projection are tuned for an image captioning task (illustrated in Figure 1). We can then evaluate its performance on vision-language (VL) tasks at test time by exploring the text the LM generates. Because of the simplicity of the linear transformation, we would expect that if the conceptual representation spaces of the two models are structured similarly, this transfer will be successful and the LM will have little trouble describing the contents of images.

We use three different image encoders with increasing levels of linguistic supervision in pretraining: BEIT (Bao et al., 2021), Normalizer Free Resnet50 (NFRN50) (Brock et al., 2021), and CLIP (Radford et al., 2021) to train different projections into the LM. By *linguistic supervision*, we refer to the extent to which the image encoder was exposed to language data during its pretraining, thus influencing the expected representational similarity between it and an LM. While CLIP was pretrained to align images with full natural language captions in a shared image-text representation space, BEIT had no exposure to language and was trained by predicting the contents of masked out sections of images. NFRN50 falls somewhere between these extremes: having been pretrained on an image classification task for identifying the subject of an image over the set of classes in ImageNet1k Russakovsky et al. (2015). Although there is no natural language in this task, the pretraining objective encourages the model to map visual features along lexical categorical concepts (the image classes) derived from the WordNet hierarchy (Miller, 1995).

We show that prompting an LM with any of the three image encoders effectively transfers semantic content in the image that the LM describes with natural language. However, performance also appears proportional to the strength of the linguistic supervision the image encoder had. While CLIP and NFRN50 perform competitively with tuning the models freely (e.g., Tsimpoukelli et al. (2021), Eichenberg et al. (2021)), BEIT appears to transfer mostly coarse-grained visual *properties* and struggles with encouraging the LM to generate exact lexical *categories*. We interpret this as evidence that models trained on either language or vision data learn conceptual spaces that are structurally similar to each other, but that the exact degree of similarity depends on the type of supervision the image encoder receives. In summary, we show: (1) that visual semantic information can be linearly mapped to language models in the form of soft prompts without tuning any model parameters. (2) That this mapping allows generative models to describe images and answer questions about images at a level that is comparable to what is achieved by multimodal models which tune image and language representations jointly. And (3) by training our prompting pipeline with different image encoder backbones, we demonstrate that linguistic supervision in pretraining plays a key role in concept formation in models and thus, the transferability of visual features from vision to text spaces.

## 2  RELATED WORK

Our approach takes inspiration from recent work in adapting pretrained language models for accepting representations of images as inputs. Particularly, the Frozen and MAGMA models (Tsimpoukelli et al., 2021; Eichenberg et al., 2021), as well as Sung et al. (2022); Alayrac et al. (2022); Mokady et al. (2021); Luo et al. (2022); Lin et al. (2021); Zhai et al. (2022), which show that pretrained image and text networks can be tuned together on an image captioning task and applied to downstream vision-language (VL) tasks. These approaches either fine-tune the pretrained models, or train non-linear MLP projection/fusion networks between modalities, making interpretation of the representations difficult compared to our approach. Scialom et al. (2020) show a learned linear transformation is sufficient for BERT to encode image region representations which are then fed to a text decoder to generate questions about the image, but it is not well understood what abstractions LMs are able to transfer from a transformation of this type, or if a text decoder can operate on linear transformations of visual encodings directly.

Pretrained/from scratch LMs have typically been used in the past for image captioning applications in which an image representation is fed into the LM as input (Desai & Johnson, 2021; Shen et al., 2021; Devlin et al., 2015). Gui et al. (2022); Yuan et al. pretrain vision-language models from scratch using image-caption data. Zeng et al. (2022); Xie et al. (2022); Wang et al. (2022) augment multimodal performance by feeding text prompts derived from VL models into an LM, in order to incorporate knowledge learned by LM training. These show LMs can interface with visual inputs described in text; our work questions whether the visual input can be fed directly into the LM, without bridging through language first. The success of aforementioned models on VL tasks indicates there is a representational similarity learned by text and image models independently, which we investigate in this paper.

Our work is also highly related to the idea of model "stitching" (Lenc & Vedaldi, 2015) in which two different models are attached at a certain layer. `LiMBeR` can be described as stitching the output of an image encoder to the input of an LM in the form of soft prompts (Lester et al., 2021). Stitching offers distinct advantages in evaluating the representational similarity between two models, as described in Bansal et al. (2021), over other conventional methods like RSA and CKA (Kriegeskorte et al., 2008; Kornblith et al., 2019). For example, `LiMBeR` allows us to show not just that CLIP encodings are *more similar* to text encodings than BEIT representations, but that BEIT representations are nevertheless able to transfer *visual property* information to the LM (§5.3).

There has been considerable interest in recent work in establishing if LMs model aspects of the non-linguistic world in order to model language. Lu et al. (2021) show that the weights of a pretrained LM can generalize to tasks with different modalities. Hao et al. (2022) similarly show that LMs can act as interfaces for multiple modalities. Li et al. (2021) show that models of entities and situations can be derived from contextual word representations. Patel & Pavlick (2022) show that very large LMs (GPT-3 scale (Brown et al., 2020)) can learn in-context non-linguistic conceptual domains depicted in text. Our work differs from these in that we have an LM interface directly with non-text data without changing model weights and show that, although fundamentally different, the representation space of a text-only LM shares non-trivial similarities to that of several vision-based models.

## 3 METHOD: LINEARLY MAPPING FROM IMAGE TO TEXT REPRESENTATIONS

While previous work has shown success in mapping images to language model soft prompts as a method for multimodal pretraining (e.g., Frozen, Magma; see Section 2), there have been no attempts to restrict the mechanism behind this mapping and understand how it works. Our basic approach is to train a single linear layer $P$ to project from the hidden size $h_I$ of a pretrained image encoder into the input space $e_L$ of a generative language model for an image captioning task. The projected inputs do not correspond to discrete language tokens, and can be thought of as soft prompts (Lester et al., 2021) representing the image. For brevity, we refer to training $P$ as **Li**nearly **M**apping **Be**tween **R**epresentation spaces (i.e., `LiMBeR`)[1]. Our approach can also be viewed as paring down the method used in Tsimpoukelli et al. (2021) and Eichenberg et al. (2021), such that the only trained parameters reside in the projection $P$. By freezing the image encoder $E$ and LM on either side of the projection, we can examine the similarities between the representation spaces of the two as a function of the ability of the LM to describe an image input or perform some task relating to it. We expect that, if a language model represents visual conceptual information structurally similarly to that learned by a vision encoder, then a simple linear transformation to the language space is all that is required to transfer visual features into the language model. Before describing the training procedure, we will describe the basic components of the model, and the variations we chose.

**Language Model** $LM$ **& Image Encoders** $E$  We hypothesize that the conceptual representations learned by an LM are equivalent, up to a linear transformation, to the representations from an image encoder $E$. The language model used is the 6 billion parameter decoder-only GPT-J model (Wang & Komatsuzaki, 2021). $P$ is trained to project from $h_I$ to the input space $e_L = 4096$ of the LM. We train several models with different $E$'s to determine the compatibility between encodings from

---

[1]We avoid specifying images or text in our backronym because one could linearly map between any two representation spaces of any modalities (e.g. video-to-text or text-to-text)

$E$ and the LM. We also test how the choice of $E$ influences performance on this task, specifically, with regards to the degree of linguistic supervision $E$ saw in pretraining, as described in Section 1.

From $E$ we extract an image encoding of dimensionality $h_I$ representing the image. We then project that encoding to a $e_L * k$ sequence of soft prompts, which we hereafter refer to as *image prompts*. $k$ is determined by the architecture of the $E$. For example, for consistency with the MAGMA model, we use the 12x12x3072d feature map before pooling from CLIP, which we flatten to $k = 12 * 12 = 144$. The encoders we experiment with are (1) **CLIP RN50x16** (Radford et al., 2021), $k = 144$, $h_I = 3072$. Because CLIP is trained to learn multimodal image-text embeddings, we expect that it will be easier for the model to learn a projection into language space than a vision only encoder. (2) **NFRN50** (Brock et al., 2021), $k = 2$, $h_I = 2048$. We train three variants using NF-Resnet50: one pretrained and frozen during caption training (NFRN50), one tuned during caption training (NFRN50 Tuned; note that the LM is still frozen), and one randomly initialized (NFRN50 Random). The NFRN50 models are pretrained on an image classification task on data that is labeled according to WordNet hypo/hypernym structure. This signal trains the model to separate object classes according to these words. For this reason, we consider it to have indirect access to linguistic supervision. (3) **BEIT-Large** (Bao et al., 2021), $k = 196$, $h_I = 1024$. BEIT is pretrained using a self-supervised masked visual token modeling task and does not have access to any labeled data which may give the model an inductive bias towards a linguistic structure. We use the 16-pixel patch version that was pretrained only on ImageNet22k. We additionally test two variants of this model: BEIT Random, is randomly initialized, and BEIT FT, which was pretrained on the same task and then finetuned for image classification on the same dataset. We use this model to show that it is indeed the linguistic supervision of the pretraining objective which induces better performance in the captioning task.

## 3.1 TRAINING PROCEDURE

Following the MAGMA and Frozen models (Eichenberg et al., 2021; Tsimpoukelli et al., 2021), we train a projection on an image captioning task so that we can learn to align the representation spaces of $E$ and the LM. All models are trained with the same basic hyperparameters and settings as described in the MAGMA paper (see Appendix A for details) on the Conceptual Captions 3M dataset (CC3M, Sharma et al. (2018)) for 15,000 training steps.

**Baselines** As baselines, we use NFRN50 Random, NFRN50 Tuned, and train our own instance of MAGMA$_{base}$. Please note that NFRN50 Tuned is a stand-in for the Frozen model: it is architecturally the same, but differs in that we use the hyperparameters used to train the MAGMA model. NFRN50 Random allows us to test the efficacy of `LiMBeR` when the image encoder backbone has not learned any useful visual features. The MAGMA we train uses the CLIP RN50x16 image encoder (Radford et al., 2021), GPT-J as the LM, and adapters in sequence in the attention blocks with a downsample factor of 4.

## 3.2 LIMITATIONS

Due to computational constraints, we did not control for the prompt length ($k$) for each image encoder. Tsimpoukelli et al. (2021) experiment with a small range for the value of $k$ for the Frozen model and show that while there are some differences, $k$ is mostly a factor in hyperparameter tuning and should not strongly affect the comparison between models. We use much higher values of $k$ for CLIP and BEIT, and this is therefore is not strongly controlled for in our study.

We consider *LM runoff* another potential confound. In some cases, if the LM recognizes and generates a relevant word for one concept (e.g., "the beach"), it might continue generating relevant information due to a strong linguistic prior for that info showing up (e.g., "building a sandcastle"), giving the illusion it is recognizing every element in an image (even if it never saw "the sandcastle"). Regardless, the scope of this problem is very limited, and across multiple large datasets our results show that recovery of *any* image information is still possible, even if the full and precise extent of which is impossible to know. We also include a 'blind' model in visual question answering analysis to further control for this.

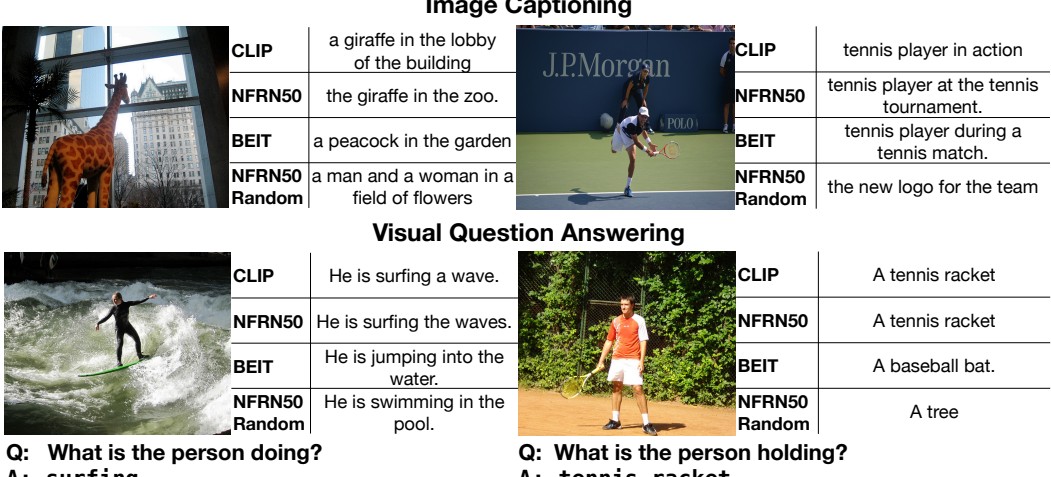

Figure 2: Curated examples of captioning and zero-shot VQA illustrating the ability of each model to transfer information to the LM without tuning either model. We use these examples to also illustrate common failure modes for BEIT prompts of sometimes generating incorrect but conceptually related captions/answers.

## 4  PERFORMANCE ON VISION-LANGUAGE TASKS

We first verify that image representations that are linearly projected into the input space of the LM carry semantic information about the content of the image that the LM can make sense of. Since we only tune a single projection between the image encoder and text decoder, the prompt tokens in the LM are equivalent to the image representation up to that linear transformation. If LMs are learning a conceptual space that reflects that of the non-linguistic, purely visually grounded space of the image encoder, the LM should be able to capture the image information and describe it in text.

**Data** We evaluate on image prompts generated by each image encoder on multiple image captioning datasets: MSCOCO (Lin et al., 2014) and NoCaps (Agrawal et al., 2019), as well as the VQA2 (Goyal et al., 2017) visual question-answering dataset. Following convention from *SimVLM* and MAGMA, we input the prefix "A picture of" after every image to prompt the model. Like in previous work, we find that this is a favorable prompt which tends to increase performance.

**Metrics** For image captioning, we report CIDEr-D (Vedantam et al., 2015), CLIPScore, and Ref-CLIPScore (Hessel et al., 2021). CIDEr-D rewards generating accurate words which are more likely to be visually informative, and CLIPScore can evaluate similarity between an image and caption without references, which helps us give credit for captions that vary greatly from the ground truth, but similar in semantic content (e.g. describing a pool as a lake). We report additional captioning metrics in Appendix B. For visual question answering, we follow the few-shot procedure used in Eichenberg et al. (2021) in which we prompt the models with the "[image] Q: [q] A:" format. We take the first word of the generation and, like in the MAGMA paper, truncate to the length of the longest ground truth answer. We also use the normalization procedure and accuracy metric described in the VQA repo[2]

**Results** Our main results can be seen in Table 1. As evidenced by comparing MAGMA and CLIP, and NFRN50 tuned and frozen, we find that there is relatively little benefit in training parameters in either encoder or decoder. Note that the MAGMA model we implemented is identical to the frozen CLIP model, with the exceptions that MAGMA tunes the image encoder and LM. On captioning and VQA tasks, performance of the jointly-tuned models (MAGMA, NFRN50 Tuned) is not consistently better, and is often worse, than just training the projection with frozen models. This trend persists across over 10 automatic captioning metrics, which are described in Appendix B. Our results indicate that there is in fact a relationship between the linguistic supervision of the pretraining task

---

[2]https://github.com/GT-Vision-Lab/VQA

| Image Captioning | NoCaps - CIDEr-D | | | | NoCaps (All) | | CoCo | CoCo | |
|---|---|---|---|---|---|---|---|---|---|
| | In | Out | Near | All | CLIP-S | Ref-S | CIDEr-D | CLIP-S | Ref-S |
| 🔥NFRN50 Tuned | 20.9 | 30.8 | 25.3 | 27.3 | 66.5 | 72.5 | 35.3 | 69.7 | 74.8 |
| 🔥MAGMA (released) | 18.0 | 12.7 | 18.4 | 16.9 | 63.2 | 68.8 | **52.1** | 76.7 | 79.4 |
| 🔥MAGMA (ours) | **30.4** | **43.4** | **36.7** | **38.7** | 74.3 | 78.7 | 47.5 | 75.3 | **79.6** |
| ❄️BEIT Random | 5.5 | 3.6 | 4.1 | 4.4 | 46.8 | 55.1 | 5.2 | 48.8 | 56.2 |
| ❄️NFRN50 Random | 5.4 | 4.0 | 4.9 | 5.0 | 47.5 | 55.7 | 4.8 | 49.5 | 57.1 |
| ❄️BEIT | 20.3 | 16.3 | 18.9 | 18.9 | 62.0 | 69.1 | 22.3 | 63.6 | 70.0 |
| ❄️NFRN50 | 21.3 | 31.2 | 26.9 | 28.5 | 65.6 | 71.8 | 36.2 | 68.9 | 74.1 |
| ❄️BEIT FT. | **38.5** | **48.8** | **43.1** | **45.3** | 73.0 | 78.1 | 51.0 | 74.2 | 78.9 |
| ❄️CLIP | 34.3 | 48.4 | 41.6 | 43.9 | **74.7** | **79.4** | **54.9** | **76.2** | **80.4** |
| | | | | | | | | | |
| **VQA n-shots** | 0 | | | | 1 | | **2** | 4 | |
| Blind | 20.60 | | | | 35.11 | | 36.17 | 36.99 | |
| 🔥NFRN50 Tuned | 27.15 | | | | 37.47 | | 38.48 | 39.18 | |
| 🔥MAGMA (ours) | 24.62 | | | | 39.27 | | 40.58 | 41.51 | |
| 🔥MAGMA (reported) | 32.7 | | | | 40.2 | | **42.5** | 43.8 | |
| ❄️NFRN50 Random | 25.34 | | | | 36.15 | | 36.79 | 37.43 | |
| ❄️BEIT | 24.92 | | | | 34.35 | | 34.70 | 31.72 | |
| ❄️NFRN50 | 27.63 | | | | 37.51 | | 38.58 | 39.17 | |
| ❄️CLIP | 33.33 | | | | 39.93 | | **40.82** | 40.34 | |

Table 1: Captioning Performance and Visual Question Answering (VQA) accuracy for all variations on model architecture and image encoders used. On captioning, we see a consistent increasing trend in performance that correlates with an increase in linguistic supervision. However BEIT (the only vision-only model), performs far above a randomly initialized NFRN50 model and is on par with the other models on CLIPScore (CLIP-S) and RefCLIP Score (Ref-S) (Hessel et al., 2021). We see that BEIT performs at the level of our random baselines on VQA, suggesting there is a deficiency in relating visual information to more complex visual-linguistic reasoning tasks

and performance on transferring to the LM. That is, CLIP outperforms NFRN50, which outperforms BEIT. To confirm this, we apply `LiMBeR` to a BEIT model finetuned on image classification (BEIT FT.), and find that this model improves performance drastically, even outperforming CLIP on No-Caps, and improving over BEIT on all metrics, including CLIP-Score by 9-10 points. This suggests the importance of the linguistic supervision in the pretraining task, rather than perhaps architecture, as the important factor for successful transfer.

Notably, we find that even vanilla BEIT, which has no linguistic supervision in pretraining, still transfers well to the LM for captioning, far outperforming random NFRN50 across the board, which had no pretraining to learn visual features. We do find that BEIT captions using vaguer language, and/or semantically related-but-incorrect descriptions of objects (Figure 2; more examples in Appendix B). We see this reflected in the CLIPScores of the captions as well, which reward semantic similarity rather than precise lexical overlap with a reference caption. BEIT captions score 62 and 63.6 for NoCaps and COCO respectively; on average only 4.5 points behind NFRN50 but 14.3 ahead of random NFRN50. Perhaps we see the greatest failure of BEIT prompts in the inability to transfer details that the LM can use to answer questions about images (At 4-shot VQA, BEIT scores 31.72% while a 'blind' LM with no image input scores 36.99%). We hypothesize this is because BEIT representations do not encode visual information that corresponds well to lexical categories. In Section 5, we provide evidence in favor of this hypothesis, and investigate the granularity of detail prompts from each frozen encoder transfer to the LM.

## 5 TRANSFER OF VISUAL CONCEPTS

Examining the conditions that cause an image prompt to succeed or fail to transfer to the LM can help us understand the differences between the text and image representation spaces. Doing so can also help us understand why BEIT prompts perform so poorly for VQA despite performing decently for captioning. In Section 5.1, we analyze the ability to accurately generate specific lexical categories in captions when they appear in images (e.g., mentioning "squirrel" when given a picture of one).

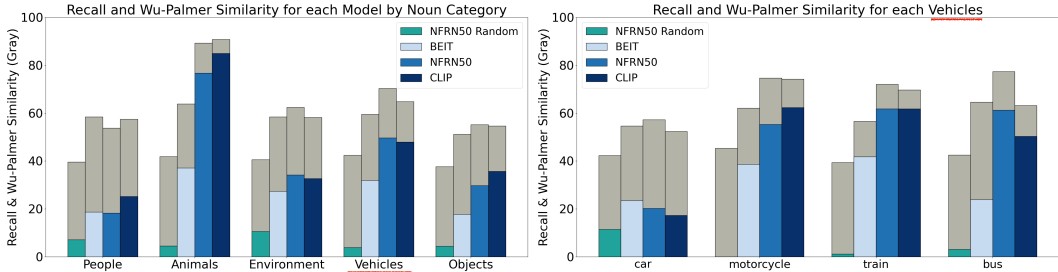

Figure 3: On average, recall of nouns in generated captions follows the standard pattern (CLIP>NFRN50>BEIT). However, judging by Wu-Palmer similarity, BEIT performs nearly the same or better than NFRN50 and CLIP on 4/5 of the noun categories. This indicates that although BEIT struggles to transfer the exact correct concept, it is transferring a related one based on visual similarity. On the right we show this effect for individual vehicle words. BEIT may have never learned to distinguish the 'bus' concept, but the LM still understands to generate a highly related concept, i.e., another vehicle. Average random Wu-Palmer similarity is around .4 consistently.

Following that, in Section 5.3 we focus on mistakes the models make: when the LM generates a bad caption, does it generate a caption that describes entities with similar visual *properties*? For example, a caption generated from an image of a "small", "woodland", and "furry" animal might not mention the actual animal depicted (e.g., a squirrel); but does it instead mention a different but similar furry animal (e.g., a rabbit)? We find that only linguistically informed image encoders (NFRN50, CLIP) tend to strongly encode concepts aligning to lexical *categories*, but all pretrained models including BEIT encode property information approximately equally well, and far better than a randomly initialized image encoder baseline.

## 5.1 TRANSFER OF LEXICAL CATEGORICAL CONCEPTS

Using the COCO validation set, we count the top 50 nouns, modifiers (e.g., adjectives), and relations (e.g., verbs, prepositional phrases) that appear in the ground truth captions and calculate how often they appear in the generated captions that were used to calculate the scores in Table 1.

**Metrics** We calculate the precision/recall/F1 for each word, broken down along conceptual categories. To test our hypothesis that BEIT transfers coarser information, we also report the Wu-Palmer similarity (Wup) (Wu & Palmer, 1994) between the ground truth word and the most similar word in the generated caption. The Wup score works by calculating the distance between the ground truth word and the generated word in the WordNet taxonomy, offering a way to measure 'how close' a word was to the correct answer.

**Results** In Figure 3, we show that BEIT's recall for nouns in categories like 'people', 'environment', 'vehicles', and 'objects' is lower than NFRN50 or CLIP, but is comparable in terms of Wup similarity in many categories. Unlike NFRN50 and CLIP's pretraining, BEIT's pretraining does not encourage it to learn conceptual differences between two similar looking objects that use different words. Compared to prompts from a randomly initialized NFRN50, for which very few consistent patterns emerge, the LM can still extract the broad conceptual meaning behind BEIT prompts, as evidenced by high Wup similarity (and CLIPScore results in Table 1). We interpret these results as supporting the hypothesis that BEIT prompts transfer conceptual information from the purely visual to purely text space, but only in terms of coarse-grained conceptual information corresponding to visual properties, not lexical categories. Our full analysis, including additional metrics and results for each individual word from the top 50 nouns, modifiers, and relations can be found in Appendix B.

## 5.2 PROBING

To rule out the possibility that BEIT representations *are* encoding lexical concept information, but are merely unable to linearly transfer it to the LM due to representational differences, we train linear probes on several datasets for image classification. We find that BEIT typically does not encode fine-

grained information as well as NFRN50 or CLIP, though it far outperforms the randomly initialized NFRN50 baseline. We discuss training details and results in Appendix E.

## 5.3 TRANSFER OF COARSE-GRAINED PERCEPTUAL CONCEPTS

To better understand what BEIT encodes, if not word category information, we further investigate where errors arise, and how the structures of the embedding spaces for each frozen image encoder differ. For the sake of this analysis, we constrain the task to generating captions for pictures of animals. The reason for this narrower scope is that the captions are easier to analyze: the caption describing a picture of an animal should virtually always mention the name of that animal, and the word used to describe the animal is mostly unambiguous.

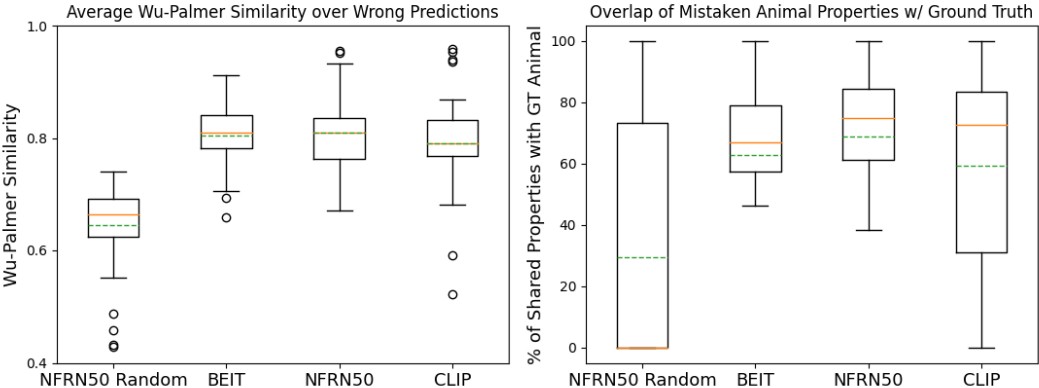

(a) **Left**: Wu-Palmer Similarity for captions in which the models don't mention the animal show that BEIT, NFRN50, and CLIP are all similarly close, meaning that even if they predict the wrong animal, it is on average very taxonomically similar. **Right**: When the model mistakes one animal for another in the dataset, how similar are the AWA properties for the true animal and the one it mistakes it most for? The average number of overlapping properties show that animals predicted from BEIT are at least as similar to the real animal as NFRN50 and CLIP. Median is shown as the solid orange line while the dashed green line shows the mean.

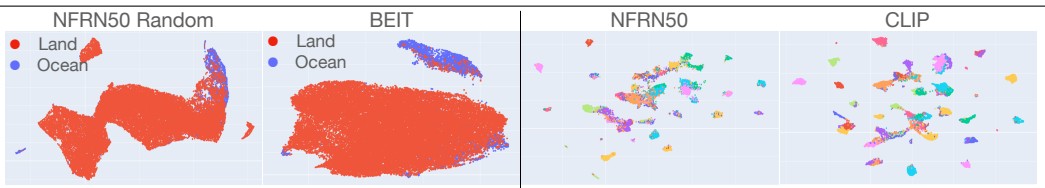

(b) **UMAP projections of AWA images:** While NFRN50 and CLIP cluster tightly along lexical categories (color coded by animal), BEIT clusters the most distinctly along animals that live in water/the ocean; the randomly initialized NFRN50 mostly randomly overlap in one cluster.

Figure 4

**Data** For this task we use the Animals With Attributes 2 (AWA) dataset (Xian et al., 2019) which contains  37k total images covering 50 animal classes. Each animal class also comes with annotations for 85 properties describing the animals (e.g., 'claws', 'stripes', 'jungle'), which allow us to analyze if prompts from certain encoders consistently make mistakes along any of these dimensions.

**Metrics** When an image prompt produces a caption, we can measure the similarity of any animals mentioned to the WordNet synset of the ground truth animal label. We can also measure similarity using the annotated properties provided by the AWA dataset. For a given animal (e.g., "squirrel"), we can look at the other animal in the dataset that it is most often mistaken for (e.g., "rabbit") and compare the proportion of properties that they share.

**Results** We generate captions for each image using prompts from each frozen image encoder. We consider a caption to be 'correct' if it contains the name of the animal the image depicts. CLIP and NFRN50 are correct most often: 59% and 43% of the time respectively. BEIT and the randomly initialized NFRN50 only achieve 13% and 0.4% accuracy, respectively. This aligns with previous

observations that BEIT struggles with encoding fine-grained lexical level concepts. By looking at failure cases for each model, we can establish whether each model is predicting the presence of a similar animal or not. In Figure 4a, we show that when captions generated from each model mistake one animal for another, the mistaken animals are highly similar to the ground truth animal when measuring both Wu-Palmer similarity (Averages: BEIT: 0.8, NFRN50: 0.81, CLIP: 0.8) and overlap of AWA properties (Averages: BEIT: 0.62, NFRN50: 0.68, CLIP: 0.59). Although BEIT prompts do not transfer the exact animal concept to the LM, the coarse grained perceptual information is transferred and 'understood' by the LM. In Figure 4b we create UMAP projections of the encodings for each image in AWA and indeed find that NFRN50 and CLIP cluster according to tight lexical categories (the animal types), BEIT clusters most tightly by perceptual features, such as habitat, having flippers, etc.

## 6 DISCUSSION & FUTURE WORK

We connect image representations as inputs to an LM with an extremely simple transformation: a linear projection. We interpret the success of generating text relevant to an image with this approach as indicative of an underexplored representational similarity between language and vision representations. Depending on the linguistic guidance in the pretraining task used to pretrain the image encoder, we see varying performance. Using a vision-only encoder (BEIT) leads to generations from the LM that are often incorrect but close under measures of perceptual relatedness. Unless finetuned with image classification BEIT has no inductive bias in pretraining to distinguish concepts that we might normally distinguish with language, especially when they are perceptually very similar. The fact that only image encoders trained with linguistic supervision can do this suggests interesting future work on the role of language in category formation. Despite strong performance with a linear transformation, the representation spaces of these models seem to contain differences that cannot be approximated in the language space. Multimodal models, ideally, will learn richer representations by taking advantage of these differences. It is useful to think about how current multimodal pretraining objectives succeed and/or fail at doing this. `LiMBeR` can serve as a strong baseline for future multimodal models, as it provides a point of comparison for a minimal mapping between vision and language.

We see `LiMBeR` as a useful tool for understanding how representations trained from different modalities can be similar or different. For concepts that are represented similarly, can we take advantage of this fact to reduce the amount of data required for learning good text representations? Where they are different, can multimodal models learn richer representations by incorporating information from both? For example, vision data may help with reporting bias in text corpora (Paik et al., 2021). Answering these questions can help us understand the limits of text-only pretraining, as well as how to better ground LMs to non-linguistic data.

## 7 CONCLUSION

In this paper, we test how similar pretrained image and text representations are by training a linear map to project image representations into the input space of a language model (LM), such that the LM can accurately describe the contents of the images. We show that models trained through `LiMBeR` (Linearly Mapping Between Representation spaces) are competitive on image captioning and visual question answering benchmarks with similar models like MAGMA that tune both image and text networks. However, we also find that such transfer is highly dependant on the amount of *linguistic supervision* the image encoder backbone had during its pretraining phase. BEIT, which is a vision-only image encoder underperforms compared to a Resnet model trained on image classification, which in turn underperforms compared to CLIP, which was pretrained with natural language captions. We explore what conceptual information transfers successfully, and find through analysis of generated text, clustering, and probing that the representational similarity between LMs and vision-only image representations is mostly restricted to coarse-grained concepts of perceptual features, while linguistically supervised vision models can transfer lexical concepts. Our findings indicate that LMs and vision models learn conceptually similar representation spaces, such that a minimal linear transformation is an adequate approximation for transferring information about an image. The extent of this representational similarity is not well understood and is an interesting direction for future work.

## 8 REPRODUCIBILITY STATEMENT

We are committed to making all of our results reproducible. Code for training all models, as well as weights of the linear projections can be found here: `https://github.com/jmerullo/limber`. We also release the weights of the linear projections that were trained for `LiMBeR`. Because we froze the image and text models attached to the projection, the weights can be used to quickly reproduce our results with the corresponding off-the-shelf pretrained models with no other tuning necessary. We use the default data splits for all datasets we used and release the random seeds used for all tasks that require generation from the LM in our codebase as well. For the AWA tasks that require matching Wordnet synsets, we document the exact animal synsets that we used as the 'ground truth' for the animal labels in Appendix D, Table 6.

## 9 ACKNOWLEDGMENTS

We would like to thank Aaron Traylor and Nihal Nayak for thoughtful discussions and feedback on this work, as well as StabilityAI for donating compute resources for model training. This research is supported in part by ODNI and IARPA via the BETTER program (2019-19051600004). The views and conclusions contained herein are those of the authors and should not be interpreted as necessarily representing the official policies, either expressed or implied, of ODNI, IARPA, or the U.S. Government.

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

## A  LiMBeR Training Details

### A.1  Training Details

We mimic the MAGMA pretraining process for each of our models as outlined in Eichenberg et al. (2021). As described above, the training task is caption generation. For each training example, an image and caption pair $(x, y)$ is fed into the model. The image encoder $E$ encodes the image into a $i_1, ..., i_k$ of dimensionality $h_I$ and length $k$. For the CLIP Encoder, for example, we extract (12,12,3072) feature patches, which we resize to (144,3072) and feed into the projection layer $P$. The output of $P$ is fed as tokens representing the image into the language model $LM$. The caption $y$ is tokenized as $t_1, ..., t_m$ where $m$ is the variable length of the caption. $LM$ is given the encoded image tokens and starting with $t_1$ learns to minimize the next token log probability of $t_i$ for timestep $i$ conditioned on $i_1, ..., i_k$ and $t_1, ...t_{i-1}$.

During training, we minimize the loss with the AdamW (Loshchilov & Hutter, 2018) optimizer per mini-batch, with the help of ZeRO stage 2 (Rajbhandari et al., 2019). We use a dropout probability of 0.1, a weight decay of 0, betas = (0.9, 0.95), and gradient clipping = 1.0. All models are trained for 15,000 training steps across 16 A100 GPUs for approximately 1.75 days. Our effective batch size was 2048. We use a learning rate of $8 * 10^{-4}$ for the projection layer $P$. For models where we tune $E$ as well, we tune its parameters with a learning rate of $2 * 10^{-6}$.

| Model | Prompt Length ($k$) | Hidden size ($h_I$) | Pretraining |
|---|---|---|---|
| CLIP RN50x16 | 144 | 3072 | Contrastive Image-caption matching |
| NFRN50 | 2 | 2048 | Image Classification |
| BEIT | 196 | 1024 | Self-supervised |

Table 2: Summary of image encoders used for pretraining. Prompt length refers to the number of the tokens fed into the language model representing the image.

## B  Captioning Performance

To get a better idea of the kinds of captions LiMBeR produces, see Figure 6, which includes 15 images that were randomly selected from the COCO validation set (2017) and the generated captions for all models we test. We include a greater range captioning metric results in Table 3 for the COCO dataset. Overall, we find the same trends that we report in the main paper, which are

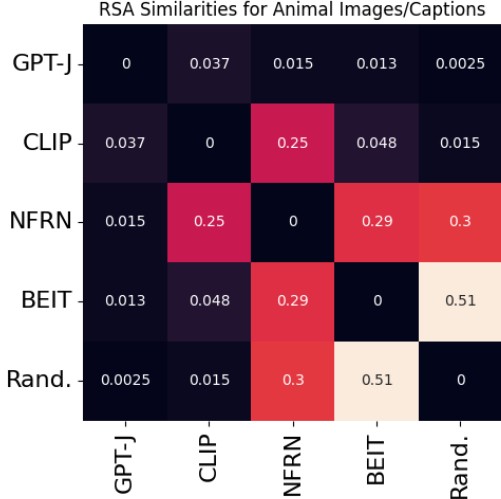

Figure 5: RSA similarity scores between representations of images of animals and the ground truth captions describing them. GPT-J is has very low correlation with image encoder representations

that 1.) the greater amount of linguistic supervision that an image encoder has, the better its captioning performance and 2.) unfreezing the image encoder does not seem to lead to consistently significant improvements. We also include the breakdown of the SPICE metric across the associated subcategories such as relations, attributes, and objects. Of the `LiMBeR` models, we find that CLIP based models do the best across the board (12.1 for CLIP vs. 9.28 for NFRN50). Besides the random baseline, BEIT performs the worst overall except in the color category (0.45 vs. 0.42 for NFRN50). We also include heatmaps with recall/precision/F1/Wu-Palmer similarity metrics comparing the captions generated by each model and the top 50 nouns (objects), modifiers, and relations from the ground truth captions from the COCO validation set (Figures 7, 8, 9).

We also use COCO images and captions to measure the similarity between vision and text encodings of the same concepts. That is, if there is a structural similarity between image and text encodings, do typical representational similarity metrics reflect this? For this experiment we want to encode a large number of images and captions that depict a set of concepts, and compare the relative representational similarity between the concepts for the image and text representations. The intuition is that if image and text models are representing concepts in a similar structure, then two concepts that are represented similarly in an image model should be similar within text representations as well. We sample a small subset of images depicting 10 different animals from the COCO dataset, and encode each image with each of our four image encoders (without a `LiMBeR` projection), and the ground truth captions with GPT-J. We use the last hidden state of the last token as the representation of the caption. We choose animals because it is easy for humans to intuitively compare how similar two animals are. In total, we collect 939 images and captions for each animal class and compare the representational similarity of the subset of encodings using the representational similarity analysis (RSA) technique from Kriegeskorte et al. (2008). The similarity matrix between each representation for each set is calculated, and then the upper triangular matrix for each similarity matrix is used to calculate the Pearson correlation. The results of this can be seen in Figure 5. We zero out the diagonal for visibility of the other values, since the diagonal is always equal to 1. RSA does not seem to capture similarity in a way that reflects the transfer performance to the LM. For example, similarity between the BEIT and randomly initialized NFRN50 model are unusually high given differences in performance in our experiments. Further analysis on the geometry of these representation spaces is required to make stronger statements about how (dis)similar they are. We leave this for future work.

## C  VISUAL QUESTION ANSWERING PERFORMANCE

The VQA accuracies are broken down by question type and model for the 4-shot case in Table 5. We don't notice any significant patterns in the per question type breakdown. It does seem perhaps

| Model | BLEU-1 | BLEU-2 | BLEU-3 | BLEU-4 | METEOR | ROUGE | CIDER | SPICE | CLIPScore | RefCLIPScore |
|---|---|---|---|---|---|---|---|---|---|---|
| NFRN50 Tuned | 0.375 | 0.229 | 0.134 | 0.080 | 0.127 | 0.316 | 0.353 | 0.091 | 0.697 | 0.748 |
| MAGMA (Ours) | 0.309 | 0.216 | 0.145 | 0.097 | 0.146 | 0.341 | 0.475 | 0.110 | 0.753 | **0.796** |
| MAGMA (Released) | **0.432** | **0.300** | **0.203** | **0.137** | **0.159** | **0.376** | **0.521** | **0.117** | **0.767** | 0.794 |
| BEIT Random | 0.288 | 0.132 | 0.052 | 0.022 | 0.068 | 0.232 | 0.052 | 0.022 | 0.488 | 0.562 |
| NFNRN Random | 0.261 | 0.115 | 0.044 | 0.018 | 0.062 | 0.208 | 0.048 | 0.021 | 0.495 | 0.571 |
| BEIT | 0.319 | 0.187 | 0.105 | 0.060 | 0.106 | 0.299 | 0.223 | 0.064 | 0.636 | 0.697 |
| NFRN50 | 0.409 | 0.254 | 0.149 | 0.088 | 0.132 | 0.334 | 0.362 | 0.093 | 0.689 | 0.741 |
| BEIT FT. | **0.420** | **0.283** | 0.182 | 0.116 | 0.155 | 0.367 | 0.510 | 0.116 | 0.742 | 0.789 |
| CLIP | 0.400 | 0.278 | **0.187** | **0.126** | **0.161** | **0.376** | 0.549 | 0.121 | 0.762 | **0.804** |

Table 3: All of the caption metrics for the models. For all scores, higher is better

| Model | SPICE (All) | Relations | Cardinality | Attributes | Size | Color | Object |
|---|---|---|---|---|---|---|---|
| NFRN50 tuned | 9.1 | 1.12 | 0.02 | 1.64 | 0.33 | 0.40 | 19.9 |
| MAGMA (ours) | 11.0 | 1.37 | 0.00 | 3.94 | 0.36 | 1.31 | 22.6 |
| MAGMA (released) | **11.7** | **1.99** | **1.30** | **3.95** | **0.50** | **1.32** | **23.8** |
| NFRN50 random | 2.14 | 0.06 | 0.00 | 0.22 | 0.00 | 0.12 | 4.93 |
| BEIT | 6.4 | 0.61 | 0.0 | 1.26 | 0.21 | 0.45 | 14.1 |
| NFRN50 | 9.28 | 1.28 | 0.02 | 1.70 | 0.27 | 0.42 | 20.2 |
| CLIP | **12.1** | **1.83** | **0.08** | **3.76** | **0.39** | **0.75** | **25.08** |

Table 4: F-scores (x100) for each fine-grained category of the SPICE metric, evaluated on the 2017 COCO validation dataset. The top and bottom divide separates models where the image encoder is either tuned of frozen, respectively. Models that use CLIP as their image encoder show a large jump in improvement over other models (even compared to tuned ResNet), especially in the Attributes (e.g. adjectives) and Object (e.g. nouns) categories.

NFRN50 is better at questions that require counting objects in an image. Future work is needed to determine if this is just noise or a significant trend.

| Q Type | # Q's | Blind | RN Tune | MAGMA | RN Rand | BEIT | RN | CLIP |
|---|---|---|---|---|---|---|---|---|
| how many | 20462 | 33.6 | 34.7 | 32 | 33 | 18.8 | **34.8** | 23.3 |
| is the | 17265 | 64.1 | 64.7 | **65.3** | 64.8 | 58.9 | 64.6 | 61.1 |
| what | 15897 | 17.4 | 19.9 | 22.4 | 16.7 | 8.9 | 20.3 | **26.9** |
| what color is the | 14061 | 30.5 | 33.7 | 36.7 | 33.1 | 28.9 | 34 | **40.3** |
| what is the | 11353 | 15.5 | 18.7 | 23.2 | 16.1 | 11.9 | 18.7 | **29.5** |
| none of the above | 8550 | 38.5 | **40.1** | 36 | 38.7 | 26.8 | 39.9 | 37.4 |
| is this | 7841 | 62.4 | 64.1 | **65.8** | 64.2 | 57.7 | 63.7 | 57.1 |
| is this a | 7492 | 63.3 | 63.8 | **67.6** | 63.4 | 59.5 | 63.5 | 58.5 |
| what is | 6328 | 9.3 | 12.9 | 21.2 | 10 | 6.6 | 13.2 | **24.1** |
| what kind of | 5840 | 15 | 22.5 | 29.5 | 15.8 | 11.1 | 22.5 | **36.9** |
| are the | 5264 | 63 | 64.2 | **64.3** | 63.9 | 59.4 | 63.8 | 58.9 |
| is there a | 4679 | 60.6 | 61.6 | **63.8** | 61.7 | 60.2 | 61.5 | 58.6 |
| what type of | 4040 | 16.4 | 23 | 30.7 | 17.2 | 12.8 | 23 | **38.2** |
| where is the | 3716 | 2 | 1.1 | 2.4 | 0.9 | 1.0 | 1.3 | **3.1** |
| is it | 3566 | **63.6** | 60.3 | 62.3 | 60 | 57.5 | 60.4 | 58.3 |
| what are the | 3282 | 13.4 | 17.3 | 23.1 | 14.7 | 9.8 | 18 | **28.6** |
| does the | 3183 | 66.6 | 67.6 | 66 | 67.4 | 63.0 | **67.6** | 64.6 |
| is | 3169 | 63.8 | 63.4 | **65.0** | 63.1 | 56.6 | 62.6 | 58.5 |
| is there | 3120 | 62.4 | 62.3 | **64.6** | 61.7 | 59.2 | 63.1 | 54.5 |
| what color are the | 3118 | 32 | 36.3 | 35.8 | 34.6 | 29.4 | 36 | **39.7** |
| are these | 2839 | 62.3 | 65.1 | **66.8** | 64.1 | 60.3 | 64.9 | 56.6 |
| are there | 2771 | 60.1 | 60.2 | **62.8** | 60.1 | 57.5 | 60.1 | 55.4 |
| what is the man | 2663 | 10 | 16.7 | 30.1 | 11.4 | 10.3 | 15.6 | **34.7** |
| is the man | 2511 | 61.9 | 61.7 | **64.6** | 62.2 | 58.9 | 61.5 | 59.2 |
| which | 2448 | 23.8 | 26 | **27.9** | 22.7 | 7.1 | 26.1 | 26 |
| how | 2422 | **12.2** | 10.2 | 7.5 | 9.7 | 4.0 | 10.8 | 9.8 |
| are | 2359 | 60.6 | 62.9 | **63.5** | 62.4 | 57.1 | 62.5 | 56.8 |
| does this | 2227 | 66.3 | 67.2 | 66.2 | 67.6 | 62.7 | **67.7** | 61.3 |
| what is on the | 2174 | 11.5 | 14.9 | 17.8 | 13.2 | 7.3 | 15.2 | **23.0** |
| how many people are | 2005 | 27.2 | **28.0** | 26.7 | 27 | 10.6 | 27.7 | 18.7 |
| what does the | 1970 | 7 | 7.6 | 12 | 6.8 | 2.7 | 7.4 | **14.2** |
| what time | 1746 | 15.4 | 18.9 | 13.4 | 16.6 | 16.0 | 19.1 | **22.4** |
| what is in the | 1733 | 9.8 | 13.7 | 23.8 | 10.8 | 6.8 | 13.8 | **30.1** |

| Q Type | # Q's | Blind | RN Tune | MAGMA | RN Rand | BEIT | RN | CLIP |
|---|---|---|---|---|---|---|---|---|
| what is this | 1696 | 9.6 | 21.7 | 35.2 | 9.6 | 11.6 | 21.8 | **41.3** |
| what are | 1556 | 9.2 | 16.8 | 28.6 | 10.4 | 8.7 | 17.4 | **36.0** |
| do | 1503 | 68.8 | 70.1 | 65.9 | **70.2** | 61.4 | 69.7 | 64.9 |
| why | 1438 | **3.2** | 2 | 2.8 | 1.5 | 0.6 | 1.9 | 2.8 |
| what color | 1428 | 26.9 | 28.5 | 32.1 | 28.4 | 23.8 | 27.8 | **35.0** |
| are they | 1335 | 60.8 | 61.2 | **65.8** | 62.5 | 59.8 | 61.4 | 59.4 |
| what color is | 1335 | 25.4 | 26.5 | 35.8 | 26.1 | 29.1 | 26.6 | **36.4** |
| are there any | 1330 | 59.1 | 58.3 | **61.6** | 58.7 | 54.4 | 58 | 52.7 |
| where are the | 1313 | 2.8 | 1.7 | 2.4 | 1.2 | 0.9 | 1.7 | **3.3** |
| is he | 1087 | 61.8 | 62.2 | **64.2** | 61.6 | 59.8 | 63.1 | 61.2 |
| what sport is | 1086 | 22 | 37.1 | 63.4 | 24.9 | 36.6 | 36.7 | **67.6** |
| who is | 1070 | 13.6 | 14.4 | 14.3 | 13.7 | 5.3 | **14.8** | 9.6 |
| is the woman | 992 | 63.2 | 62.2 | **66.6** | 61 | 59.3 | 61.5 | 60.9 |
| has | 946 | 64 | 65.3 | 65.5 | 65.5 | 62.7 | **66.3** | 63.5 |
| what brand | 935 | **20.0** | 16.5 | 4.4 | 17.8 | 6.3 | 17.8 | 12.4 |
| how many people are in | 905 | 23.9 | 28.3 | **29.0** | 25.8 | 15.8 | 26.9 | 17.4 |
| what is the person | 900 | 8.2 | 16.3 | 32.2 | 9.2 | 12.4 | 17.7 | **38.0** |
| is this an | 890 | 65.1 | 67.6 | **68.2** | 67.7 | 61.4 | 67.4 | 59.2 |
| can you | 872 | 59.5 | 58.9 | **62.3** | 59.9 | 58.8 | 59.6 | 59.5 |
| what is the woman | 853 | 8.5 | 15 | 28.5 | 10.2 | 6.7 | 14.6 | **32.6** |
| what animal is | 833 | 10.2 | 28.5 | 56.1 | 10.8 | 16.8 | 28.6 | **63.6** |
| what is the color of the | 826 | 35.4 | 40.4 | 39.3 | 36.8 | 34.3 | 39.3 | **44.7** |
| was | 818 | 62.6 | 61.8 | **63.8** | 62.1 | 59.9 | 60.9 | 59.5 |
| is the person | 794 | 61.9 | 61.7 | 61.3 | **62.4** | 57.5 | 62 | 57.1 |
| what is the name | 780 | 3.4 | 4.4 | 8 | 3.8 | 1.5 | 4.5 | **9.0** |
| what room is | 762 | 15 | 43.4 | 62.9 | 19.9 | 24.0 | 39.3 | **65.9** |
| is this person | 734 | 62 | 61.3 | **63.3** | 62.5 | 60.2 | 61.9 | 57.5 |
| do you | 724 | 61.7 | **62.1** | 60.1 | 60.8 | 55.6 | 62 | 58.3 |
| is that a | 714 | 60.9 | **63.1** | 62.7 | 61.5 | 57.5 | 62.4 | 59.6 |
| what number is | 673 | 8.8 | 7.6 | **12.6** | 7.5 | 0.5 | 8.9 | 10.2 |
| could | 618 | 72.3 | 71.8 | 67.5 | **73.6** | 67.4 | 71.5 | 58.9 |
| why is the | 514 | 1.6 | 1.9 | 2.5 | 1.3 | 0.5 | 1.6 | **2.6** |

Table 5: Average 4-shot accuracies of models on every question type from the VQA2.0 dataset. Note that NFRN50, NFRN Random, and NFRN Tuned are renamed to save space. MAGMA refers to our version of the model.

## D  ANIMALS WITH ATTRIBUTES

**AWA2**  The Animals with Attributes 2 Dataset contains 37k images of 50 animal classes annotated with 85 properties (e.g. 'stripes', 'tough'). For each image encoder we generate captions using image projections into the LM for each image in the dataset and identify any animals mentioned using overlapping WordNet synsets with the ground truth labels. We report the animal synsets that we used as the 'ground truth' synsets. For example, to calculate accuracy, etc. we split generated captions on white space and check the possible synsets of each word and check if there is any overlap with our list. Each label synset is listed in Table 6.

We report two AWA experiments here: (1) the first appears in the main paper, and investigates how similar the most similar words mentioned in the generated captions are to the ground truth animal in terms of Wu-Palmer similarity *for the generated captions in which the ground truth animal does not appear* – i.e., mistakes. The per animal results can be found in Table 7. (2) We also report the average precision (AP) of the average animal properties vector for the animals predicted to the ground truth for each model. This experiment is similar to the properties-overlap experiment reported in the paper but is a measure of how often the predicted animal properties are similar vs. very different from the ground truth. If the LM mentions an animal in the dataset, we add that animal's property vector to a running list. If the LM does not mention an animal in the dataset, we ignore that caption. For each image-caption pair, we take the average of the properties vectors that correspond to the animals mentioned in the captions, and calculate the AP against the ground truth binary properties vector. Results per animal class are in Table 8. We exclude any entries for which a model makes fewer than 50 mistakes for an animal class. For example, for all of the images labeled as depicting a tiger in the AWA dataset, CLIP based captions fail to mention a tiger a *single* time.

| Animal Name | WordNet Synset |
| --- | --- |
| antelope | antelope.n.01 |
| grizzly+bear | grizzly.n.01 |
| killer+whale | killer_whale.n.01 |
| beaver | beaver.n.07 |
| dalmatian | dalmatian.n.02 |
| persian+cat | persian_cat.n.01 |
| horse | horse.n.01 |
| german+shepherd | german_shepherd.n.01 |
| blue+whale | blue_whale.n.01 |
| siamese+cat | siamese_cat.n.01 |
| skunk | skunk.n.04 |
| mole | mole.n.06 |
| tiger | tiger.n.02 |
| hippopotamus | hippopotamus.n.01 |
| leopard | leopard.n.02 |
| moose | elk.n.01 |
| spider+monkey | spider_monkey.n.01 |
| humpback+whale | humpback.n.03 |
| elephant | elephant.n.01 |
| gorilla | gorilla.n.01 |
| ox | ox.n.02 |
| fox | fox.n.01 |
| sheep | sheep.n.01 |
| seal | seal.n.09 |
| chimpanzee | chimpanzee.n.01 |
| hamster | hamster.n.01 |
| squirrel | squirrel.n.01 |
| rhinoceros | rhinoceros.n.01 |
| rabbit | rabbit.n.01 |
| bat | bat.n.01 |
| giraffe | giraffe.n.01 |
| wolf | wolf.n.01 |
| chihuahua | chihuahua.n.03 |
| rat | rat.n.01 |
| weasel | weasel.n.02 |
| otter | otter.n.02 |
| buffalo | american_bison.n.01 |
| zebra | zebra.n.01 |
| giant+panda | giant_panda.n.01 |
| deer | deer.n.01 |
| bobcat | bobcat.n.01 |
| pig | hog.n.03 |
| lion | lion.n.01 |
| mouse | mouse.n.01 |
| polar+bear | ice_bear.n.01 |
| collie | collie.n.01 |
| walrus | walrus.n.01 |
| raccoon | raccoon.n.02 |
| cow | cow.n.01 |
| dolphin | dolphin.n.02 |

Table 6: To aid with reproducibility, we report all animal synsets that were used for experiments that require disambiguating words in captions to animal classes. This allows us to correctly count a mention of "tigress" in a caption as a mention of the "tiger" animal type without relying on unreliable string matching techniques.

| | Random | NFRN50 Random | BEIT | NFRN50 | CLIP |
|---|---|---|---|---|---|
| polar+bear | 0.79 | 0.60 | 0.86 | 0.95 | 0.96 |
| buffalo | 0.73 | 0.64 | 0.84 | 0.93 | 0.95 |
| bobcat | 0.72 | 0.71 | 0.83 | 0.90 | 0.94 |
| grizzly+bear | 0.76 | 0.70 | 0.88 | 0.95 | 0.94 |
| blue+whale | 0.73 | 0.43 | 0.69 | 0.76 | 0.87 |
| dalmatian | 0.77 | 0.72 | 0.79 | 0.83 | 0.85 |
| chihuahua | 0.74 | 0.70 | 0.84 | 0.85 | 0.85 |
| leopard | 0.77 | 0.70 | 0.82 | 0.81 | 0.85 |
| collie | 0.72 | 0.70 | 0.85 | 0.85 | 0.85 |
| german+shepherd | 0.72 | 0.67 | 0.84 | 0.85 | 0.85 |
| siamese+cat | 0.75 | 0.69 | 0.85 | 0.85 | 0.84 |
| persian+cat | 0.74 | 0.67 | 0.85 | 0.85 | 0.84 |
| ox | 0.73 | 0.69 | 0.82 | 0.84 | 0.84 |
| seal | 0.77 | 0.55 | 0.77 | 0.74 | 0.82 |
| spider+monkey | 0.74 | 0.65 | 0.81 | 0.80 | 0.81 |
| antelope | 0.75 | 0.72 | 0.78 | 0.80 | 0.81 |
| skunk | 0.79 | 0.65 | 0.84 | 0.79 | 0.81 |
| mole | 0.79 | 0.64 | 0.79 | 0.84 | 0.80 |
| lion | 0.77 | 0.74 | 0.82 | 0.80 | 0.79 |
| rat | 0.80 | 0.67 | 0.79 | 0.85 | 0.78 |
| hamster | 0.80 | 0.66 | 0.79 | 0.80 | 0.78 |
| walrus | 0.78 | 0.56 | 0.72 | 0.76 | 0.78 |
| wolf | 0.80 | 0.69 | 0.85 | 0.83 | 0.77 |
| killer+whale | 0.71 | 0.46 | 0.71 | 0.73 | 0.77 |
| humpback+whale | 0.73 | 0.43 | 0.66 | 0.67 | 0.77 |
| chimpanzee | 0.72 | 0.64 | 0.85 | 0.74 | 0.76 |
| moose | 0.75 | 0.60 | 0.81 | 0.79 | 0.74 |
| fox | 0.80 | 0.72 | 0.84 | 0.84 | 0.74 |
| weasel | 0.79 | 0.68 | 0.81 | 0.83 | 0.73 |
| hippopotamus | 0.79 | 0.58 | 0.76 | 0.74 | 0.73 |
| deer | 0.77 | 0.66 | 0.81 | 0.80 | 0.73 |
| otter | 0.79 | 0.60 | 0.80 | 0.82 | 0.72 |
| pig | 0.76 | 0.67 | 0.83 | 0.75 | 0.71 |
| cow | 0.71 | 0.65 | 0.78 | 0.78 | 0.70 |
| sheep | 0.76 | 0.65 | 0.84 | 0.82 | 0.68 |
| horse | 0.75 | 0.62 | 0.77 | 0.75 | 0.59 |
| giant+panda | 0.79 | 0.70 | 0.87 | 0.90 | 0.52 |
| beaver | 0.80 | 0.55 | 0.77 | 0.76 | – |
| tiger | 0.77 | 0.74 | 0.82 | – | – |
| elephant | 0.79 | 0.64 | 0.79 | – | – |
| gorilla | 0.72 | 0.60 | 0.91 | 0.84 | – |
| squirrel | 0.80 | 0.67 | 0.80 | 0.83 | – |
| rhinoceros | 0.78 | 0.67 | 0.80 | 0.76 | – |
| rabbit | 0.76 | 0.69 | 0.82 | 0.81 | – |
| bat | 0.82 | 0.67 | 0.74 | 0.74 | – |
| giraffe | 0.78 | 0.70 | 0.79 | 0.76 | – |
| zebra | 0.76 | 0.68 | 0.78 | – | – |
| mouse | 0.79 | 0.64 | 0.75 | 0.83 | – |
| raccoon | 0.79 | 0.64 | 0.83 | 0.81 | – |
| dolphin | 0.73 | 0.49 | 0.72 | 0.68 | – |
| **Mean=** | **0.76** | **0.64** | **0.80** | **0.81** | **0.79** |

Table 7: Wu-Palmer Similarity for mistakes. Animals for which a model made fewer than 50 mistakes are dashed out

| Animal/Model | Random | NFRN50 Random | BEIT | NFRN50 | CLIP |
|---|---|---|---|---|---|
| hamster | 0.70 | 0.74 | 0.83 | 1.00 | 1.00 |
| rat | 0.79 | 0.83 | 0.81 | 0.99 | 1.00 |
| squirrel | 0.76 | 0.70 | 1.00 | 1.00 | 1.00 |
| rhinoceros | 0.48 | 0.41 | 0.81 | 0.96 | 1.00 |
| rabbit | 0.75 | 0.76 | 0.85 | 1.00 | 1.00 |
| bat | 0.59 | 0.58 | 0.58 | 0.99 | 1.00 |
| giraffe | 0.66 | 0.73 | 0.92 | 1.00 | 1.00 |
| wolf | 0.87 | 0.84 | 0.94 | 1.00 | 1.00 |
| chihuahua | 0.80 | 0.80 | 0.79 | 0.91 | 1.00 |
| zebra | 0.79 | 0.86 | 1.00 | 1.00 | 1.00 |
| chimpanzee | 0.86 | 0.84 | 0.83 | 1.00 | 1.00 |
| giant+panda | 0.71 | 0.74 | 0.76 | 1.00 | 1.00 |
| deer | 0.83 | 0.91 | 1.00 | 1.00 | 1.00 |
| bobcat | 0.72 | 0.79 | 0.80 | 1.00 | 1.00 |
| lion | 0.74 | 0.93 | 0.93 | 1.00 | 1.00 |
| mouse | 0.77 | 0.69 | 0.81 | 0.98 | 1.00 |
| polar+bear | 0.72 | 0.76 | 1.00 | 1.00 | 1.00 |
| raccoon | 0.76 | 0.77 | 0.85 | 1.00 | 1.00 |
| grizzly+bear | 0.70 | 0.68 | 1.00 | 1.00 | 1.00 |
| dolphin | 0.55 | 0.69 | 1.00 | 1.00 | 1.00 |
| tiger | 0.77 | 0.84 | 1.00 | 1.00 | 1.00 |
| leopard | 0.79 | 0.86 | 0.95 | 1.00 | 1.00 |
| horse | 0.88 | 1.00 | 1.00 | 1.00 | 1.00 |
| german+shepherd | 0.87 | 0.86 | 0.76 | 1.00 | 1.00 |
| blue+whale | 0.42 | 0.34 | 1.00 | 1.00 | 1.00 |
| siamese+cat | 0.83 | 1.00 | 1.00 | 1.00 | 1.00 |
| killer+whale | 0.42 | 0.56 | 0.88 | 1.00 | 1.00 |
| beaver | 0.68 | 0.66 | 0.77 | 1.00 | 1.00 |
| hippopotamus | 0.47 | 0.41 | 0.80 | 1.00 | 1.00 |
| persian+cat | 0.75 | 1.00 | 1.00 | 1.00 | 1.00 |
| spider+monkey | 0.79 | 0.77 | 0.99 | 1.00 | 1.00 |
| humpback+whale | 0.43 | 0.82 | 0.97 | 1.00 | 1.00 |
| elephant | 0.61 | 0.61 | 1.00 | 1.00 | 1.00 |
| gorilla | 0.76 | 0.75 | 0.89 | 1.00 | 1.00 |
| moose | 0.77 | 0.82 | 0.88 | 1.00 | 1.00 |
| cow | 0.74 | 0.82 | 1.00 | 1.00 | 1.00 |
| fox | 0.80 | 0.77 | 0.96 | 1.00 | 1.00 |
| seal | 0.57 | 0.51 | 0.95 | 1.00 | 1.00 |
| sheep | 0.59 | 0.70 | 0.98 | 1.00 | 1.00 |
| pig | 0.72 | 0.70 | 0.98 | 1.00 | 1.00 |
| otter | 0.66 | 0.72 | 0.73 | 0.76 | 1.00 |
| mole | 0.72 | 0.68 | 0.88 | 0.82 | 1.00 |
| walrus | 0.39 | 0.38 | 0.70 | 0.72 | 0.99 |
| antelope | 0.81 | 0.85 | 0.94 | 0.97 | 0.98 |
| ox | 0.74 | 0.69 | 0.89 | 0.93 | 0.95 |
| collie | 0.84 | 0.87 | 0.80 | 0.93 | 0.91 |
| weasel | 0.85 | 0.77 | 0.86 | 0.90 | 0.90 |
| buffalo | 0.64 | 0.69 | 0.91 | 0.74 | 0.85 |
| skunk | 0.70 | 0.79 | 0.83 | 0.83 | 0.84 |
| dalmatian | 0.86 | 0.86 | 0.83 | 0.72 | 0.73 |
| **Mean=** | **0.71** | **0.74** | **0.89** | **0.96** | **0.98** |

| Animal/Model | Random | NFRN50 Random | BEIT | NFRN50 | CLIP |
|---|---|---|---|---|---|

Table 8: Per animal average precision (AP) for properties of mentioned animal in captions per model for the Animals with Attributes 2 (AWA2) dataset. BEIT, which tends to do worse at captioning and question answering consistently predicts animals which share similar properties, and considerably better than randomly initialized NFRN50 and randomly selecting animals as baselines. This suggests that BEIT representations encode similar animals into broad conceptual categories (e.g. large savanna animals) which are able to linearly transfer to the LM. Without linguistic supervision, BEIT does not naturally distinguish these by the words we use to describe them, as NFRN50 and CLIP do.

| Animal/Model | NFRN50 Random | BEIT | NFRN50 | CLIP |
|---|---|---|---|---|
| giraffe | 0.00 | 0.16 | 0.96 | 1.00 |
| zebra | 0.00 | 0.77 | 1.00 | 1.00 |
| tiger | 0.00 | 0.75 | 0.98 | 1.00 |
| elephant | 0.00 | 0.37 | 0.96 | 0.99 |
| dolphin | 0.00 | 0.37 | 0.58 | 0.98 |
| rabbit | 0.00 | 0.13 | 0.85 | 0.98 |
| rhinoceros | 0.00 | 0.04 | 0.30 | 0.98 |
| squirrel | 0.00 | 0.41 | 0.88 | 0.97 |
| gorilla | 0.00 | 0.20 | 0.93 | 0.96 |
| sheep | 0.00 | 0.27 | 0.91 | 0.96 |
| raccoon | 0.00 | 0.00 | 0.73 | 0.95 |
| bat | 0.00 | 0.00 | 0.28 | 0.94 |
| horse | 0.09 | 0.85 | 0.94 | 0.92 |
| humpback+whale | 0.00 | 0.21 | 0.56 | 0.92 |
| hippopotamus | 0.00 | 0.06 | 0.59 | 0.92 |
| fox | 0.00 | 0.15 | 0.91 | 0.91 |
| cow | 0.00 | 0.41 | 0.82 | 0.90 |
| giant+panda | 0.00 | 0.00 | 0.76 | 0.90 |
| moose | 0.00 | 0.01 | 0.41 | 0.89 |
| deer | 0.02 | 0.41 | 0.79 | 0.87 |
| pig | 0.00 | 0.15 | 0.76 | 0.86 |
| leopard | 0.00 | 0.42 | 0.92 | 0.86 |
| seal | 0.00 | 0.27 | 0.73 | 0.86 |
| wolf | 0.00 | 0.03 | 0.76 | 0.85 |
| hamster | 0.00 | 0.00 | 0.69 | 0.83 |
| mouse | 0.00 | 0.01 | 0.25 | 0.82 |
| beaver | 0.00 | 0.01 | 0.49 | 0.80 |
| rat | 0.00 | 0.00 | 0.37 | 0.78 |
| chimpanzee | 0.00 | 0.00 | 0.44 | 0.76 |
| lion | 0.08 | 0.14 | 0.40 | 0.66 |
| killer+whale | 0.00 | 0.00 | 0.10 | 0.53 |
| otter | 0.00 | 0.03 | 0.06 | 0.48 |
| mole | 0.00 | 0.01 | 0.02 | 0.40 |
| walrus | 0.00 | 0.00 | 0.00 | 0.36 |
| bobcat | 0.00 | 0.00 | 0.16 | 0.33 |
| grizzly+bear | 0.00 | 0.00 | 0.06 | 0.15 |
| chihuahua | 0.00 | 0.00 | 0.04 | 0.14 |
| antelope | 0.00 | 0.00 | 0.07 | 0.05 |
| collie | 0.00 | 0.00 | 0.06 | 0.04 |
| weasel | 0.00 | 0.00 | 0.03 | 0.03 |
| buffalo | 0.00 | 0.04 | 0.02 | 0.02 |
| dalmatian | 0.00 | 0.00 | 0.00 | 0.02 |
| ox | 0.00 | 0.00 | 0.00 | 0.01 |
| skunk | 0.00 | 0.00 | 0.00 | 0.01 |
| siamese+cat | 0.00 | 0.00 | 0.00 | 0.00 |
| german+shepherd | 0.00 | 0.00 | 0.00 | 0.00 |

| Animal/Model | NFRN50 Random | BEIT | NFRN50 | CLIP |
|---|---|---|---|---|
| persian+cat | 0.00 | 0.00 | 0.00 | 0.00 |
| polar+bear | 0.00 | 0.00 | 0.00 | 0.00 |
| spider+monkey | 0.00 | 0.00 | 0.00 | 0.00 |
| blue+whale | 0.00 | 0.00 | 0.00 | 0.00 |
| **Mean=** | **0.00** | **0.13** | **0.43** | **0.59** |

Table 9: Accuracy for each model and each animal class in Animals with Attributes 2. A caption is considered correct if the animal name is mentioned in the caption for the image.

# E  PROBING VISUAL REPRESENTATIONS

We train probes on the image encodings from each of our image encoders to classify fine-grained lexical and coarse grained categorical concepts on several datasets: COCO (Lin et al., 2014), and CC3M (Sharma et al., 2018), and CIFAR-100 (Krizhevsky et al., 2009). The architecture is a single linear layer which takes an image encoding of dimension $h_I$ (see Table 2) and projects to the number of classes for the classification task. For single label classification, we use a softmax activation on the logits and train with cross entropy as the loss function. For multilabel classification tasks, we use a sigmoid activation layer on top of the logits and train with a binary cross entropy loss function. We consider a certain class predicted if the value of the class after the sigmoid is ¿0.5.

**Hyperparameters**  For simplicity, all probes are trained with the same hyperparameters (with a few exceptions for the CC3M probes): learning rate: 1e-4, optimizer: AdamW (Loshchilov & Hutter, 2018), betas=(0.9, 0.999), batch size: 48 for CC3M probes; 32 for all others, max epochs: 300 for CC3M probes; 300 for all others.

| Object Supercategory | Objects within class |
|---|---|
| accessory | *backpack, umbrella, handbag, tie, suitcase* |
| animal | *bird, cat, dog, horse, sheep, cow, elephant, bear, zebra, giraffe* |
| appliance | *microwave, oven, toaster, sink, refrigerator* |
| electronic | *tv, laptop, mouse, remote, keyboard, cell phone* |
| food | *banana, apple, sandwich, orange, broccoli, carrot, hot dog, pizza, donut, cake* |
| furniture | *chair, couch, potted plant, bed, dining table, toilet* |
| indoor | *book, clock, vase, scissors, teddy bear, hair drier, toothbrush* |
| kitchen | *bottle, wine glass, cup, fork, knife, spoon, bowl* |
| outdoor | *traffic light, fire hydrant, stop sign, parking meter, bench* |
| person | *person* |
| sports | *frisbee, skis, snowboard, sports ball, kite, baseball bat, baseball glove, skateboard, surfboard, tennis racket* |
| vehicle | *bicycle, car, motorcycle, airplane, bus, train, truck, boat* |

Table 10: All individual object labels and supercategories they fall under as annotated in the COCO dataset.

## E.1  COCO OBJECT CLASSES PROBE

The COCO dataset contains labels for 80 objects in images, typically used for object detection or segmentation tasks. Because one image can be labeled for multiple objects, we train the probe as a multi-label classification task.

In addition to having fine-grained labels for each type, COCO provides labels for each broad category that each object belongs to, called the *supercategory*. In Table 10, we show which supercategory each object label falls under.

### E.1.1 COARSE-GRAINED SUPERCATEGORY PROBES

We train a multilabel classification probe to classify the object category seen in a given image. We report F1 for each `LiMBeR` model in Figure 11. We find that BEIT does a bit worse than NFRN50 and CLIP overall, but is able to classify some categories (images with accessories, vehicles, and 'outdoor' objects) well.

### E.1.2 FINE-GRAINED OBJECT LABEL PROBES

Next, we look at the probe results for probes trained to identify individual objects by type. Our results can be found in Figure 12d. We find the same pattern emerges, and that BEIT does not seem to be significantly closer to the other pretrained models in terms of F1 on the coarse-grained vs. fine-grained patterns as we might expect. However, we do show that BEIT encodes strong, but weaker lexical concept categories than the other two models, and that the finding that BEIT transfers coarser grained information is not due to irreconcilable representational differences between BEIT space and the LM space.

### E.2 CC3M PROBES

We also *train* the same set of probes on image data from CC3M, but *evaluate* on the same validation set from COCO (i.e., the same evaluation as used in Section E.1.2). The purpose of this experiment is to create a setting for a linear probe that better matches the `LiMBeR` setup we use in the main paper. If there are concepts that the probe has no trouble with, but rarely appear in captions, that could be an indicator that the LM and the image encoder represent that concept very differently in representation space.

**Data** To align the CC3M images with the object labels in COCO, we create labels by looking for exact string matches of the object label words (e.g. "teddy bear") in CC3M captions. Of the 80 object classes, we cut out any that have fewer than 1000 images. This leaves us with 782,794 training images for the probes across 53 object classes; fewer than the CC3M dataset used to train `LiMBeR`, but far more than the previous datasets we used for probing.

**Results** Because, in this setting, we train our probes on the same distribution as the `LiMBeR` models, we compare the F1 of the probes identifying objects in images to the F1 of those concepts appearing in the generated captions. Our results can be seen in Figure 13. It appears that if the image encoder encodes the lexical concept, it generally also transfers to the LM with `LiMBeR`. Limitations of this approach are that (1) the BEIT probe appears much worse at the domain shift from CC3M to COCO (e.g., F1 for animals drops 0.3 compared to when the probes are trained on COCO) and (2) some words in the label space are often substituted for more common words in generated captions (e.g. "person" could be generated as "man", "woman", etc.). This makes it difficult to recognize cases where the probe succeeds but the transfer fails. An interesting problem for future work is better understanding which concepts are encoded in an image encoder's representations, but do not transfer well with a linear map to the LM.

### E.3 CIFAR-100 PROBES

CIFAR-100 (Krizhevsky et al., 2009) is a dataset of 60,000 32x32 images balanced across 100 object classes. Like COCO, the 100 object labels are also annotated for coarse-grained object categories including 'aquatic mammals' and 'household furniture'. For CIFAR data, we train a classifier which classifies an image for a *single* object label. Like with COCO, we train a set of probes for the fine and coarse labels. We were surprised to find that for CIFAR images, BEIT representations tended to better than NFRN50 in terms of average F1 (for the fine-grained probe, BEIT: 0.57, NFRN50: 0.47); a first for any of the experiments we ran. Given the majority of evidence shows NFRN50 encodes lexical category information stronger than BEIT, we hypothesize this is not because of BEIT encoding lexical concepts more strongly, but due to the small resolution of images: because BEIT uses visual tokens, it may be more robust to extremely blurry images, which are out of distribution for both NFRN50 and BEIT.

### E.3.1 COARSE-GRAINED PROBES

In Figure 14 we show the F1 results for the coarse-grained image labels probes trained on image encodings from each model. Each coarse-grained class in the testing set has 500 images each.

### E.3.2 FINE-GRAINED PROBES

In Figure 15d we show the per object class F1 results for the image encodings from each model. Each class in the testing set has 100 images each. Similar to the COCO probe we do not see significant intra-model changes between the coarse-grained and fine-grained probe results.

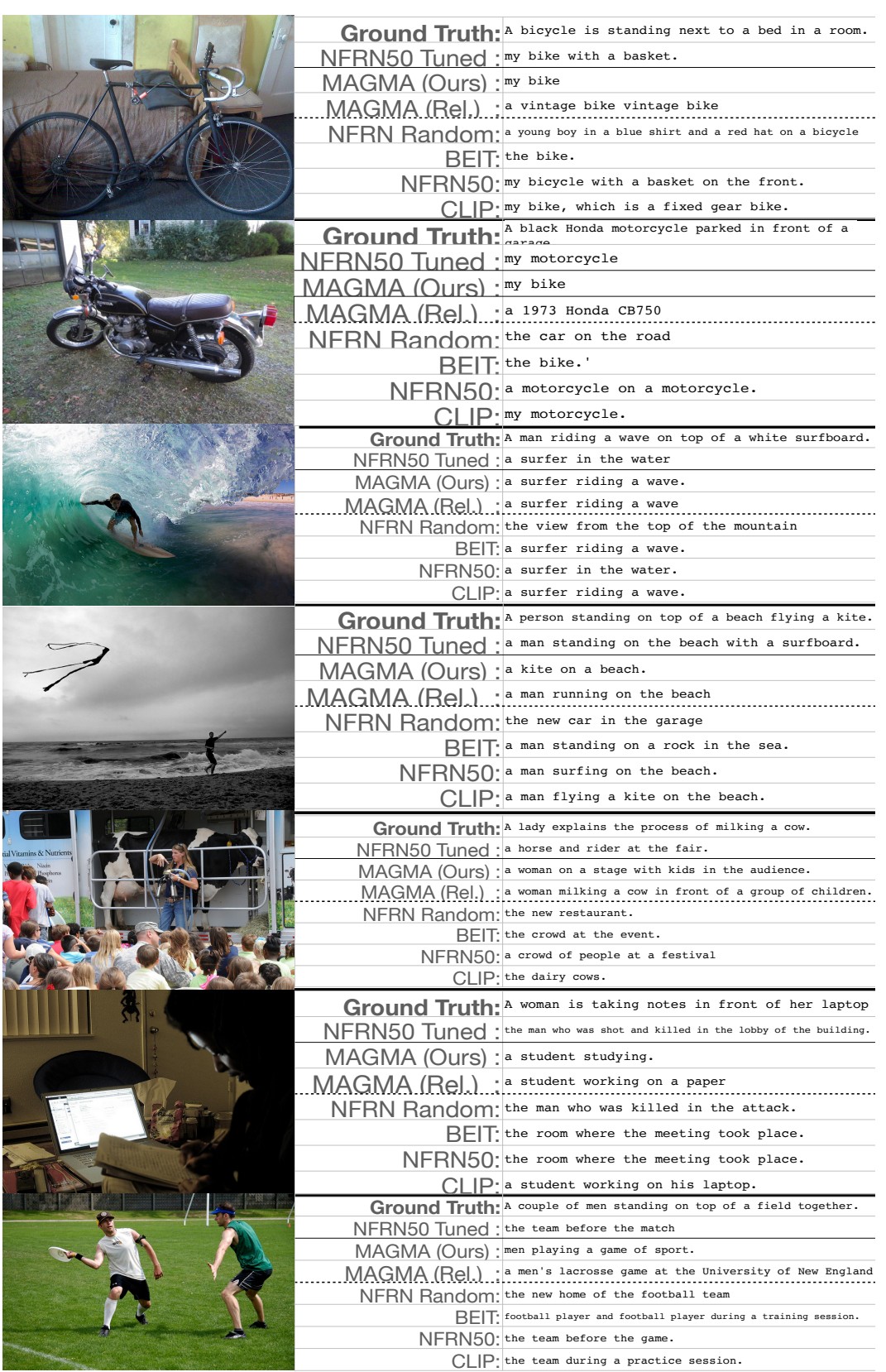

Figure 6: 15 randomly selected images from the COCO 2017 validation dataset and the generated captions from all models.

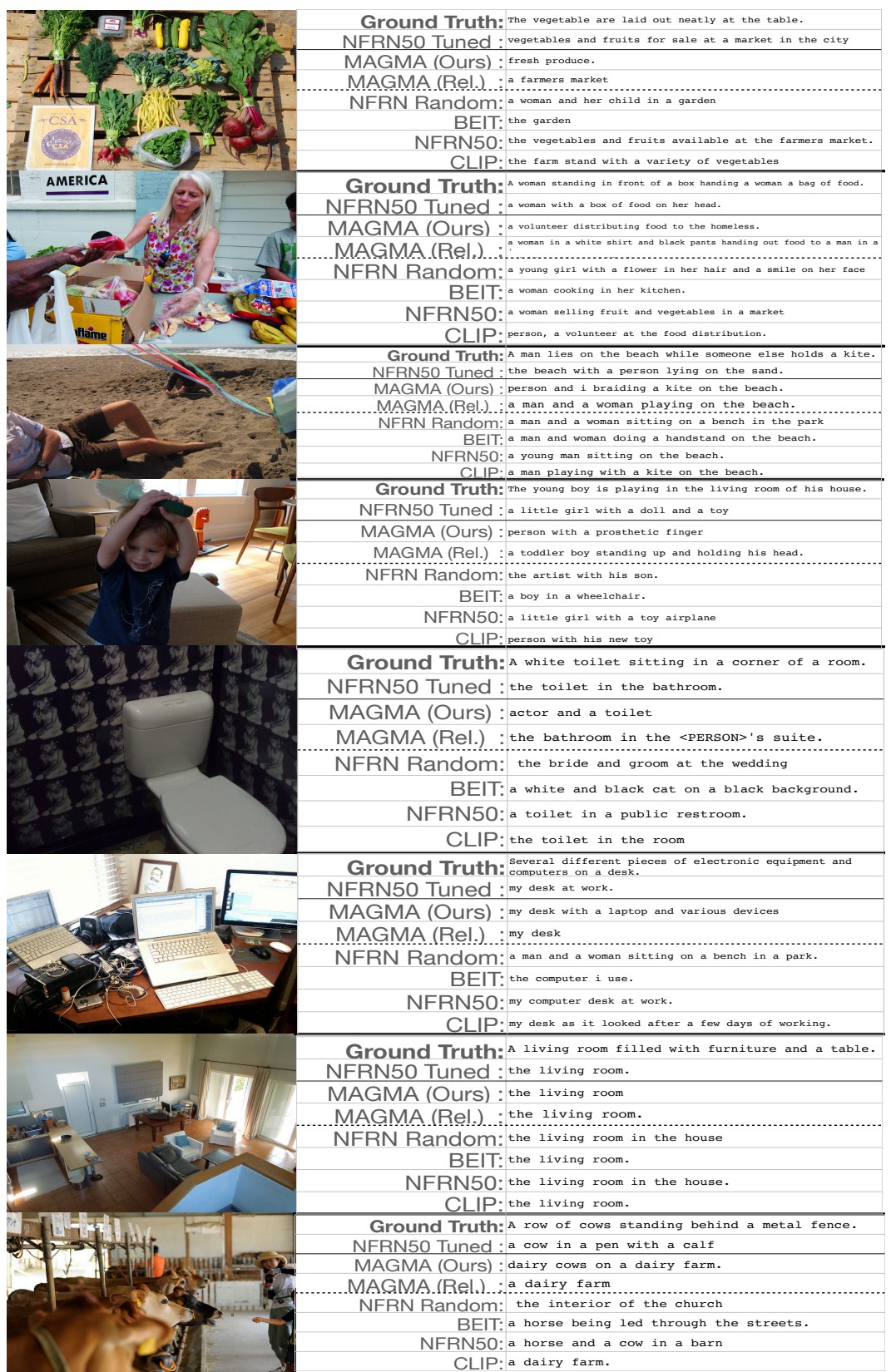

Figure 6: 15 randomly selected images from the first random seed taken from the COCO 2017 validation dataset with the generated captions from all models.

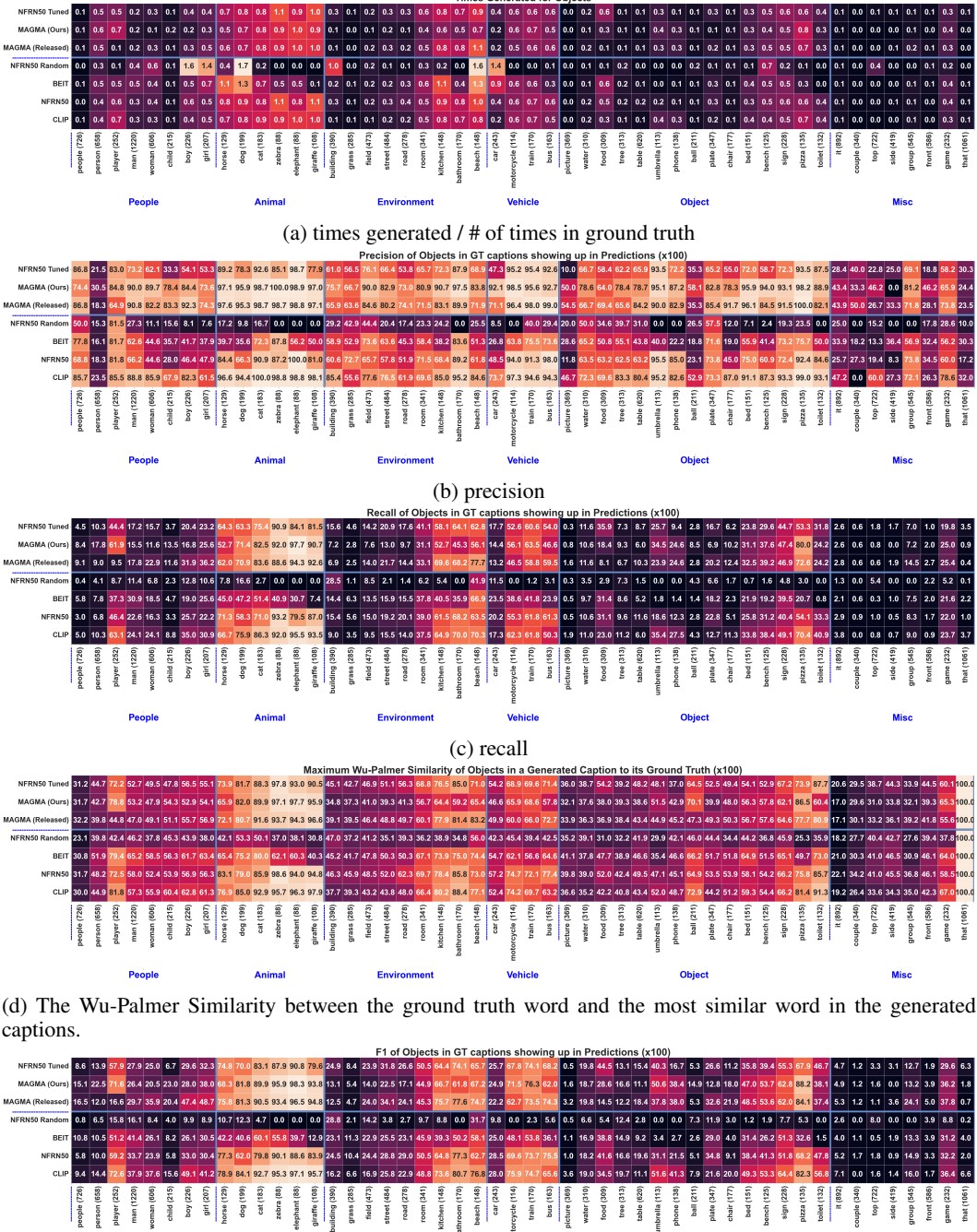

(a) times generated / # of times in ground truth

(b) precision

(c) recall

(d) The Wu-Palmer Similarity between the ground truth word and the most similar word in the generated captions.

(e) f1

Figure 7: The top 50 nouns that appear in the ground truth captions of the COCO validation set and how often each model generates them

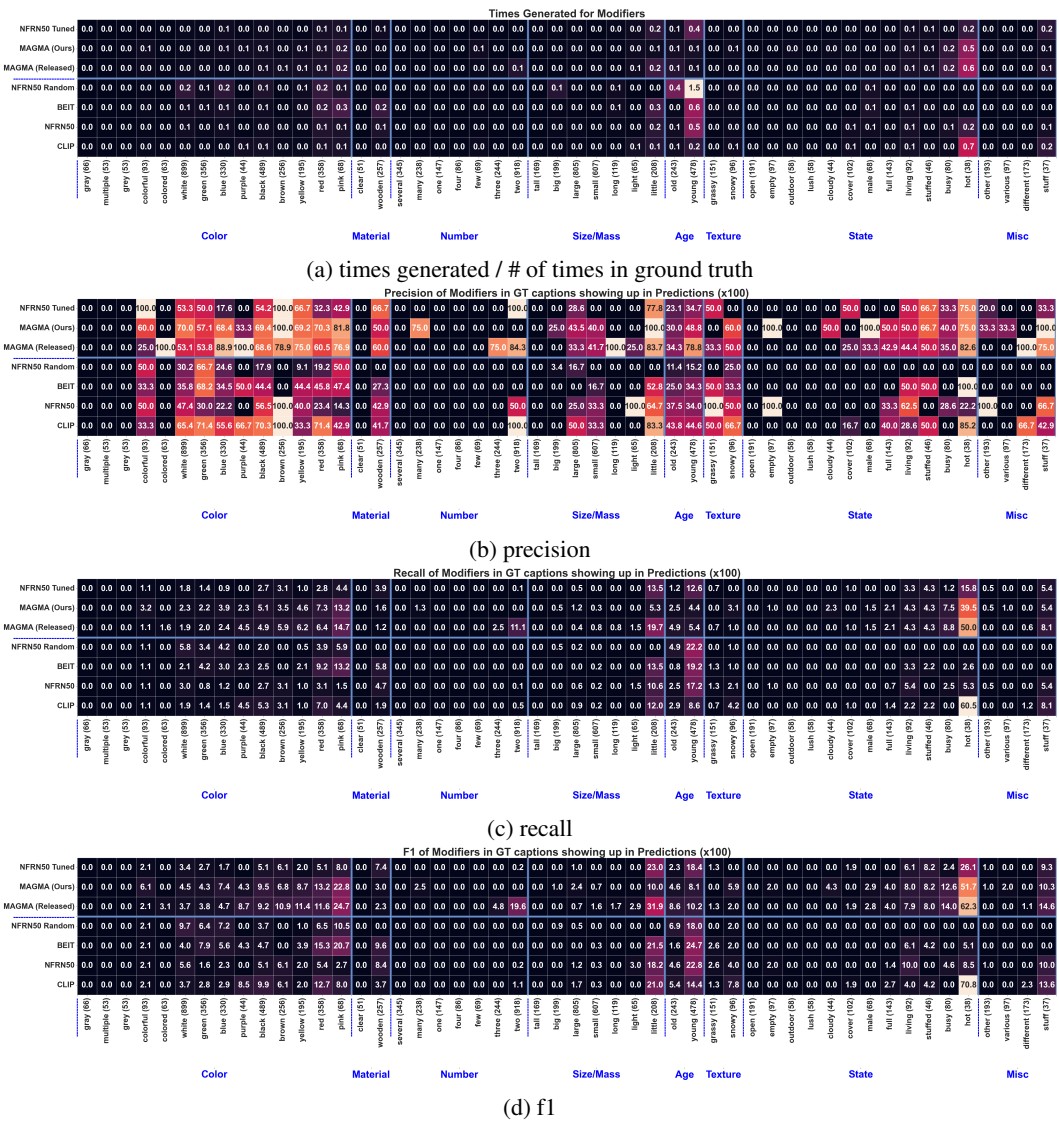

(a) times generated / # of times in ground truth

(b) precision

(c) recall

(d) f1

Figure 8: The top 50 modifiers that appear in the ground truth captions of the COCO validation set and how often each model generates them

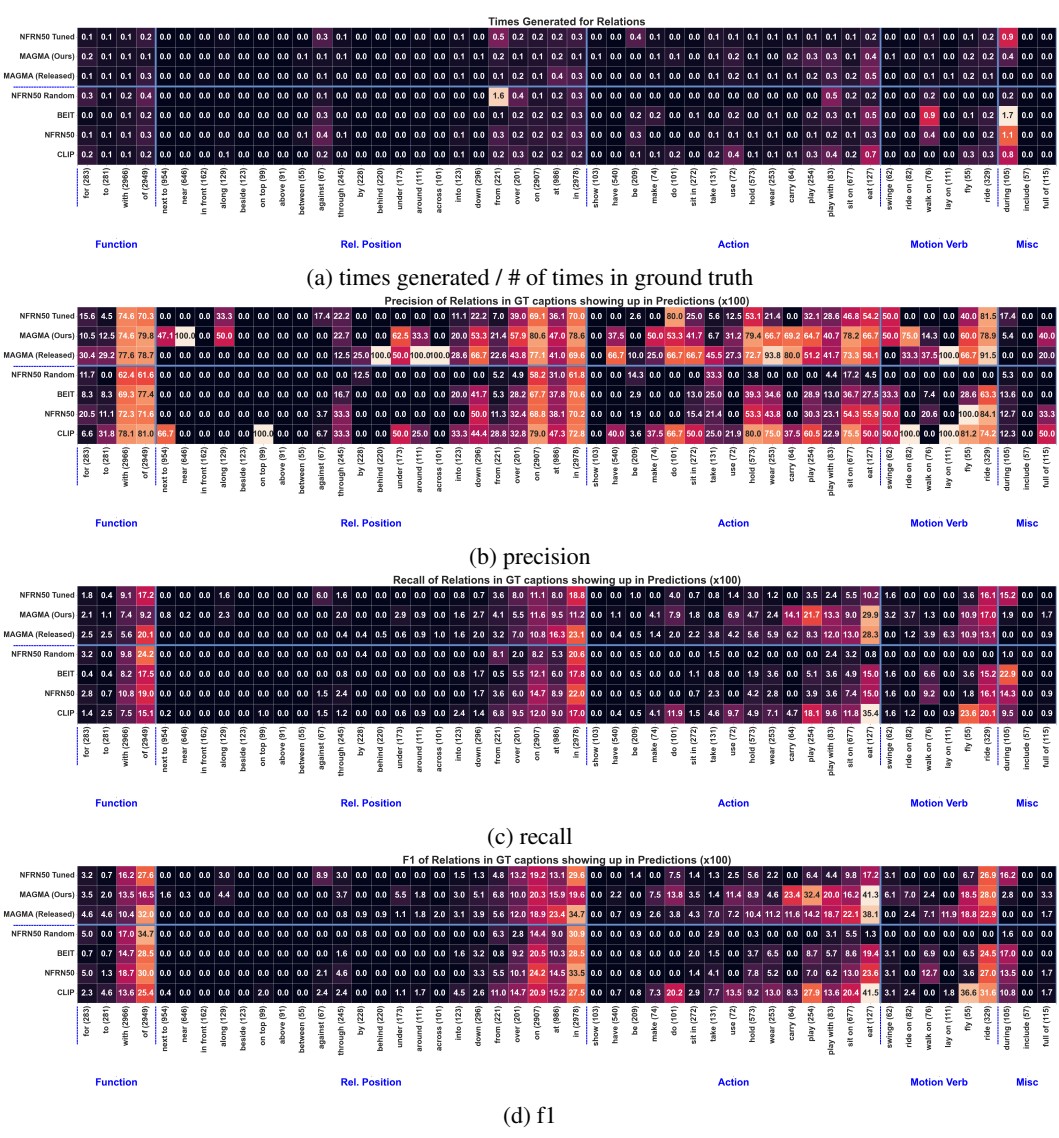

Figure 9: The top 50 relations that appear in the ground truth captions of the COCO validation set and how often each model generates them

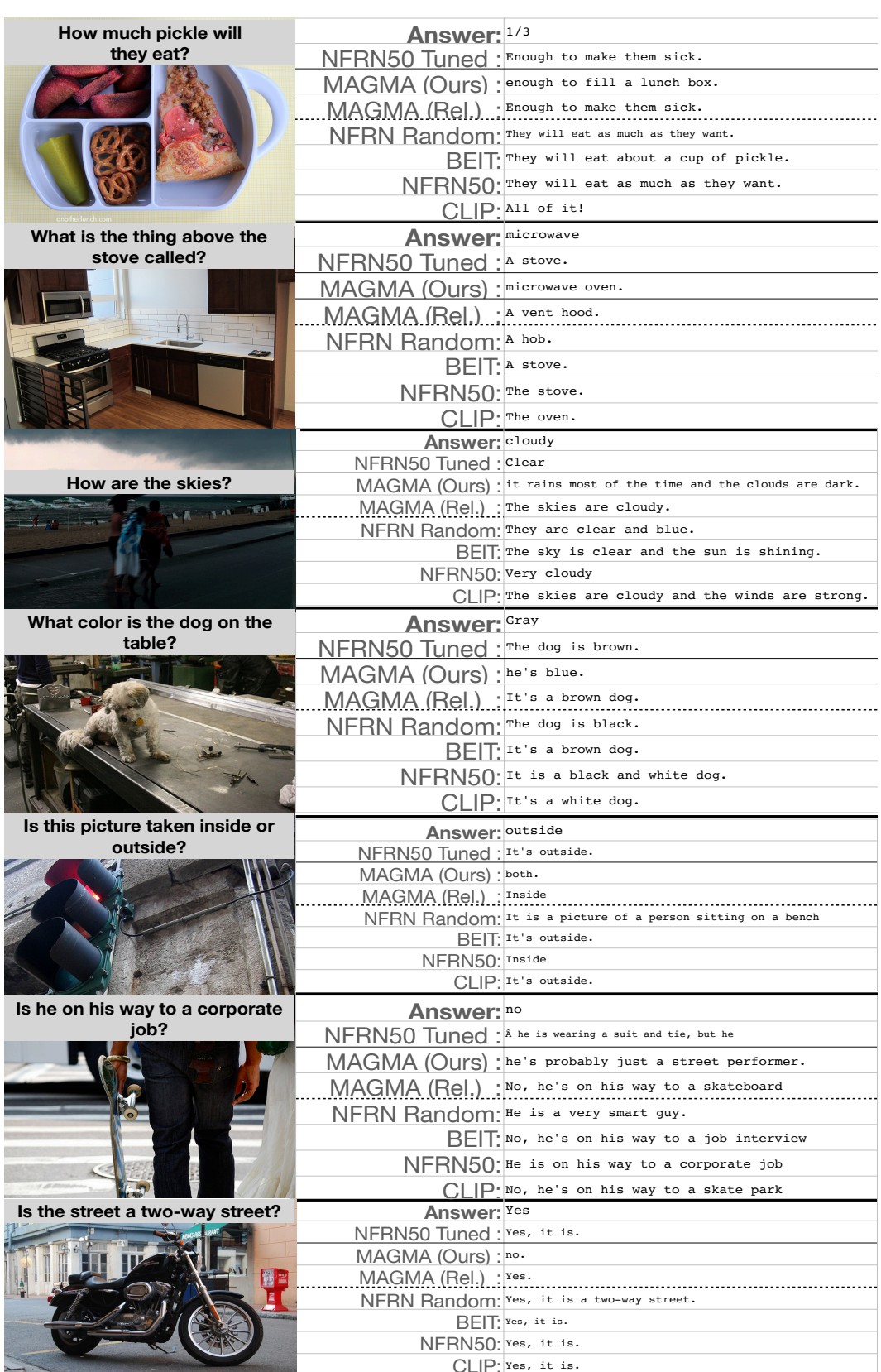

Figure 10: 15 randomly selected images from the VQA2 validation set and the generated answers from all models.

| **Does the zebra look happy?** | **Answer:** | Yes |
| --- | --- | --- |
| | NFRN50 Tuned : | No, it looks like it is about to eat you |
| | MAGMA (Ours) : | he looks like he is trying to be happy. |
| | MAGMA (Rel.) : | It's a zebra, so it's always happy |
| | NFRN Random: | Yes, he is enjoying the sunshine. |
| | BEIT: | Yes, it is happy. |
| | NFRN50: | Yes! |
| | CLIP: | Yes, he is enjoying the grass. |

| **What activity caused him to fall?** | **Answer:** | skiing |
| --- | --- | --- |
| | NFRN50 Tuned : | He was skiing on a slope that was covered in ice |
| | MAGMA (Ours) : | skiing! |
| | MAGMA (Rel.) : | He was skiing. |
| | NFRN Random: | He was a member of the wrestling team. |
| | BEIT: | He was trying to catch a ball. |
| | NFRN50: | He was skiing on a slope that was covered in ice |
| | CLIP: | He was skiing on a chair lift. |

| | **Answer:** | 60 |
| --- | --- | --- |
| | NFRN50 Tuned : | __ |
| | MAGMA (Ours) : | it took. |
| | MAGMA (Rel.) : | It took her about 10 minutes. |
| | NFRN Random: | It took her about 10 minutes to make the kite fly |
| | BEIT: | It took her about five minutes to make it fly. |
| **How many minutes did it take for the little girl to make her kite fly?** | NFRN50: | __ |
| | CLIP: | It took her about a minute to make it fly. |

| **What brand shirt is man wearing?** | **Answer:** | Polo |
| --- | --- | --- |
| | NFRN50 Tuned : | It's a brand called. |
| | MAGMA (Ours) : | brand. |
| | MAGMA (Rel.) : | <PERSON> |
| | NFRN Random: | It is a white shirt with a blue collar and a |
| | BEIT: | The shirt is a brand. |
| | NFRN50: | It's a new one. |
| | CLIP: | It's a sweater. |

| **What is the zebra doing?** | **Answer:** | eating |
| --- | --- | --- |
| | NFRN50 Tuned : | It is eating grass. |
| | MAGMA (Ours) : | he is eating grass! |
| | MAGMA (Rel.) : | It is eating grass. |
| | NFRN Random: | It is trying to get out of the water. |
| | BEIT: | It is eating grass. |
| | NFRN50: | Eating grass |
| | CLIP: | Eating grass. |

| **Is the player wearing helmet?** | **Answer:** | no |
| --- | --- | --- |
| | NFRN50 Tuned : | No, he is not. |
| | MAGMA (Ours) : | no |
| | MAGMA (Rel.) : | No. |
| | NFRN Random: | Yes, he is wearing a helmet. |
| | BEIT: | No, he is not. |
| | NFRN50: | No, he is not. |
| | CLIP: | No. |

| **What is the cat sitting on?** | **Answer:** | suitcase |
| --- | --- | --- |
| | NFRN50 Tuned : | It's a book. |
| | MAGMA (Ours) : | a suitcase. |
| | MAGMA (Rel.) : | A suitcase. |
| | NFRN Random: | It is a cat. |
| | BEIT: | It's a cat. |
| | NFRN50: | It's a laptop! |
| | CLIP: | A suitcase. |

| **Are there people in the photo?** | **Answer:** | no |
| --- | --- | --- |
| | NFRN50 Tuned : | Yes, there are people in the photo. |
| | MAGMA (Ours) : | yes |
| | MAGMA (Rel.) : | Yes, there are people in the photo. |
| | NFRN Random: | Yes, there are people in the photo. |
| | BEIT: | yes, there are people in the photo. |
| | NFRN50: | Yes, there are. |
| | CLIP: | Yes, there are people in the photo. |

Figure 10: 15 randomly selected images from the VQA2 validation set and the generated answers from all models.

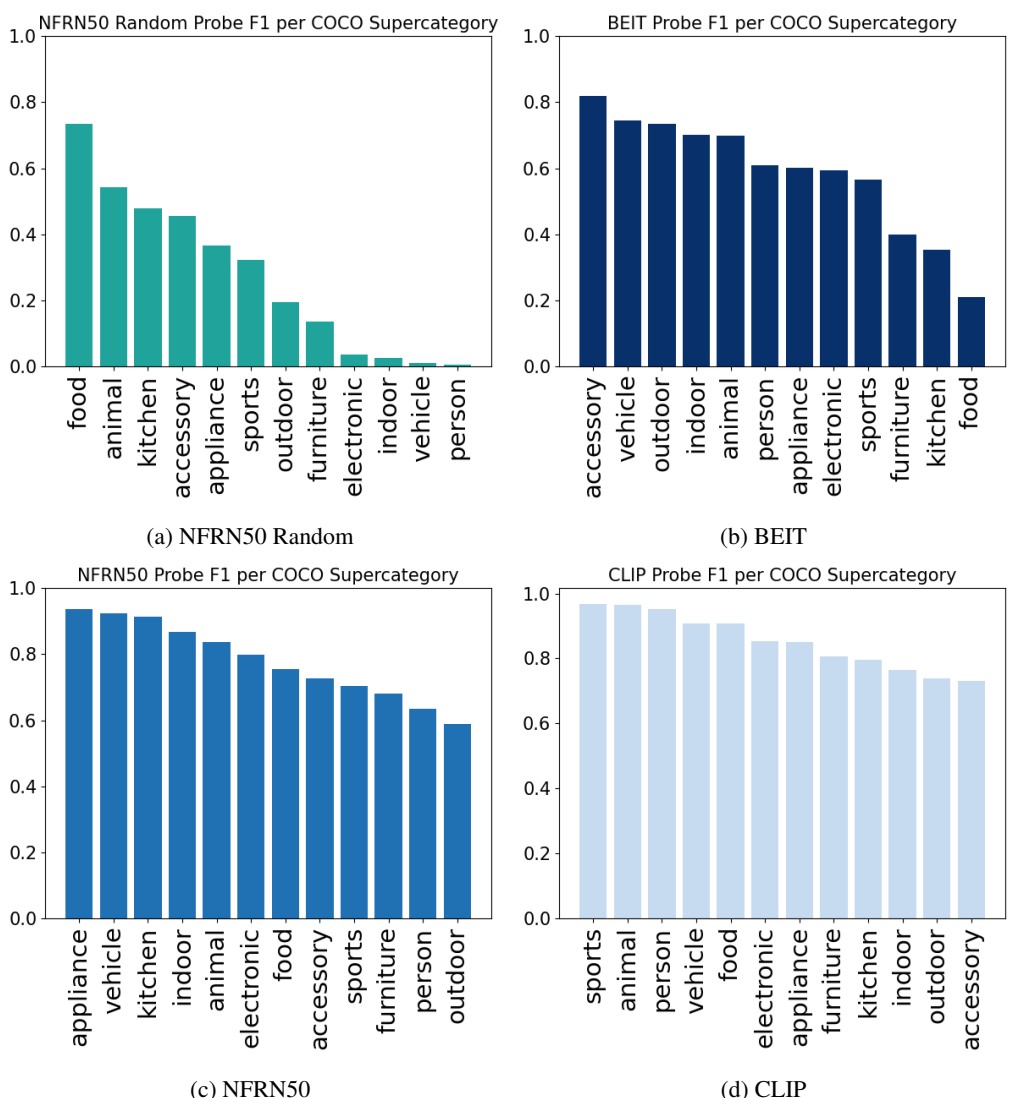

Figure 11: Probes trained on COCO images to classify the supercategories of the objects in the images

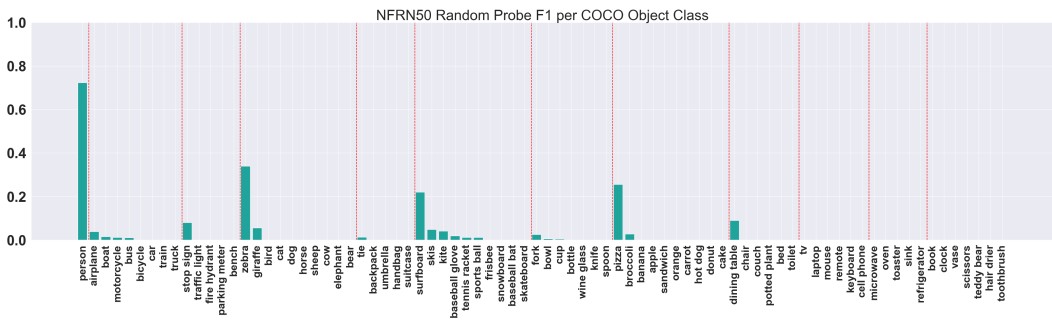

(a) NFRN50 Random Per COCO object label probe F1 results

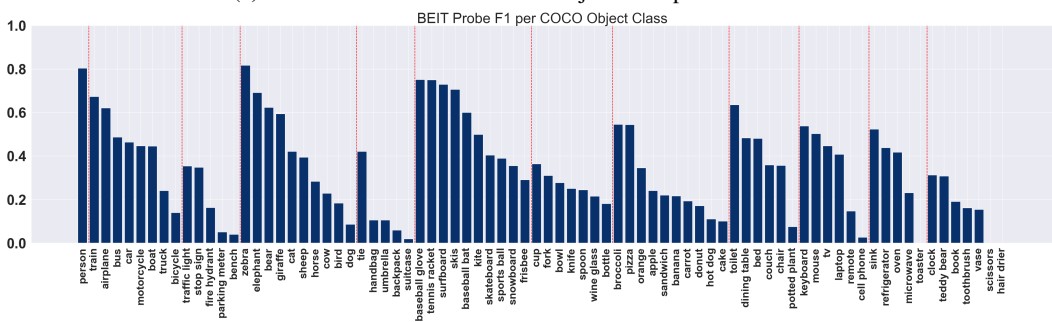

(b) BEIT Per COCO object label probe F1 results

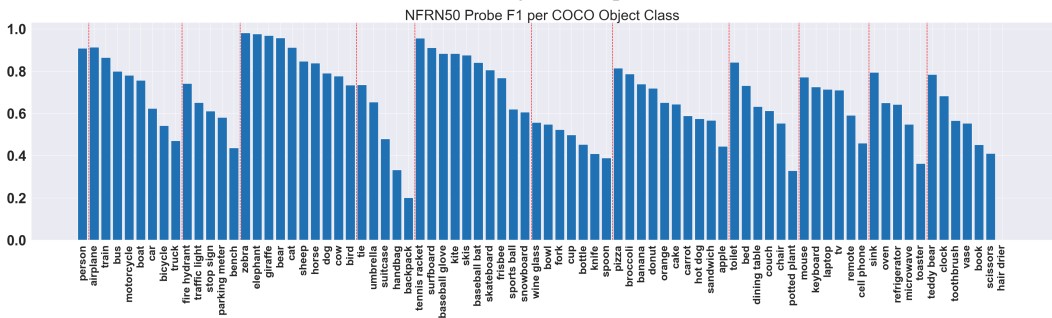

(c) NFRN50 Per COCO object label probe F1 results

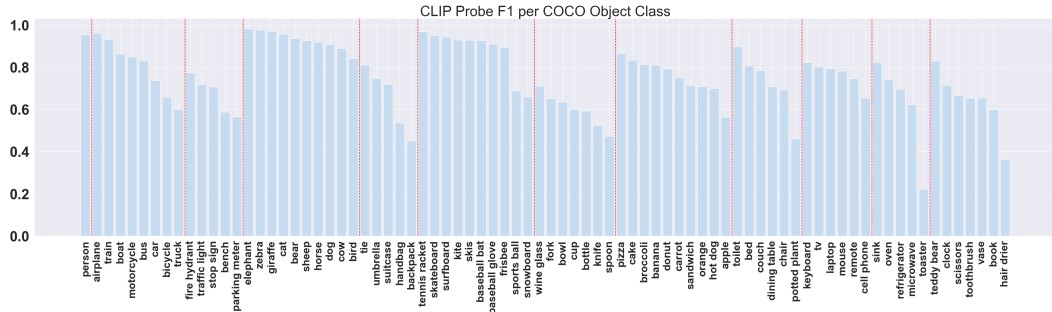

(d) CLIP Per COCO object label probe F1 results

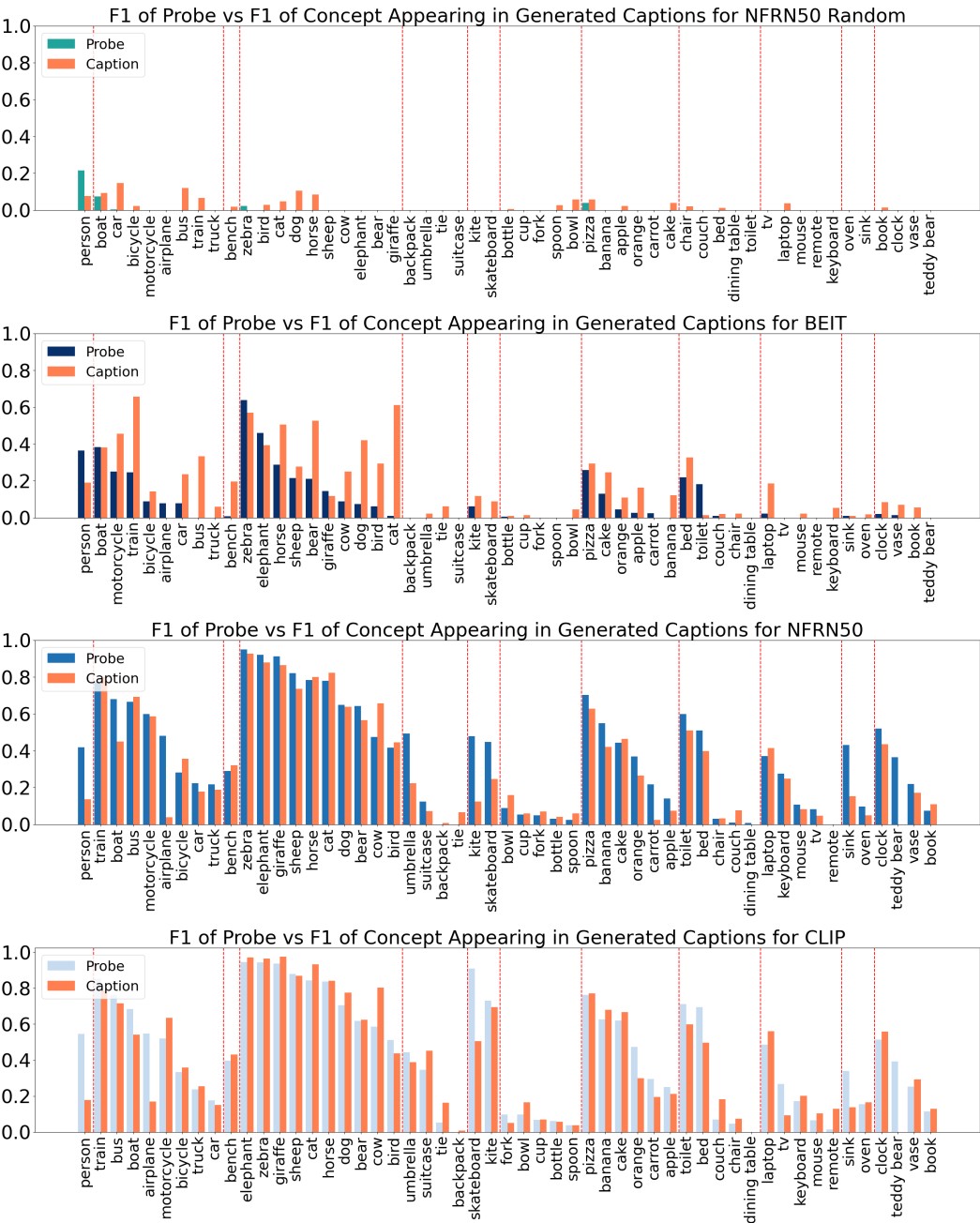

Figure 13: F1 of image encoder probes trained on CC3M and evaluated on COCO. We find that F1 of captions by object category tend to follow those of probe performance. Notably the BEIT probe is much worse at transferring from CC3M to COCO, and the captioning F1 tends to be consistently higher which makes it difficult to draw conclusions for this model. Generally, it appears the ability to *encode* lexical information into the image representation entails being able to *transfer* that information to the LM with a linear map.

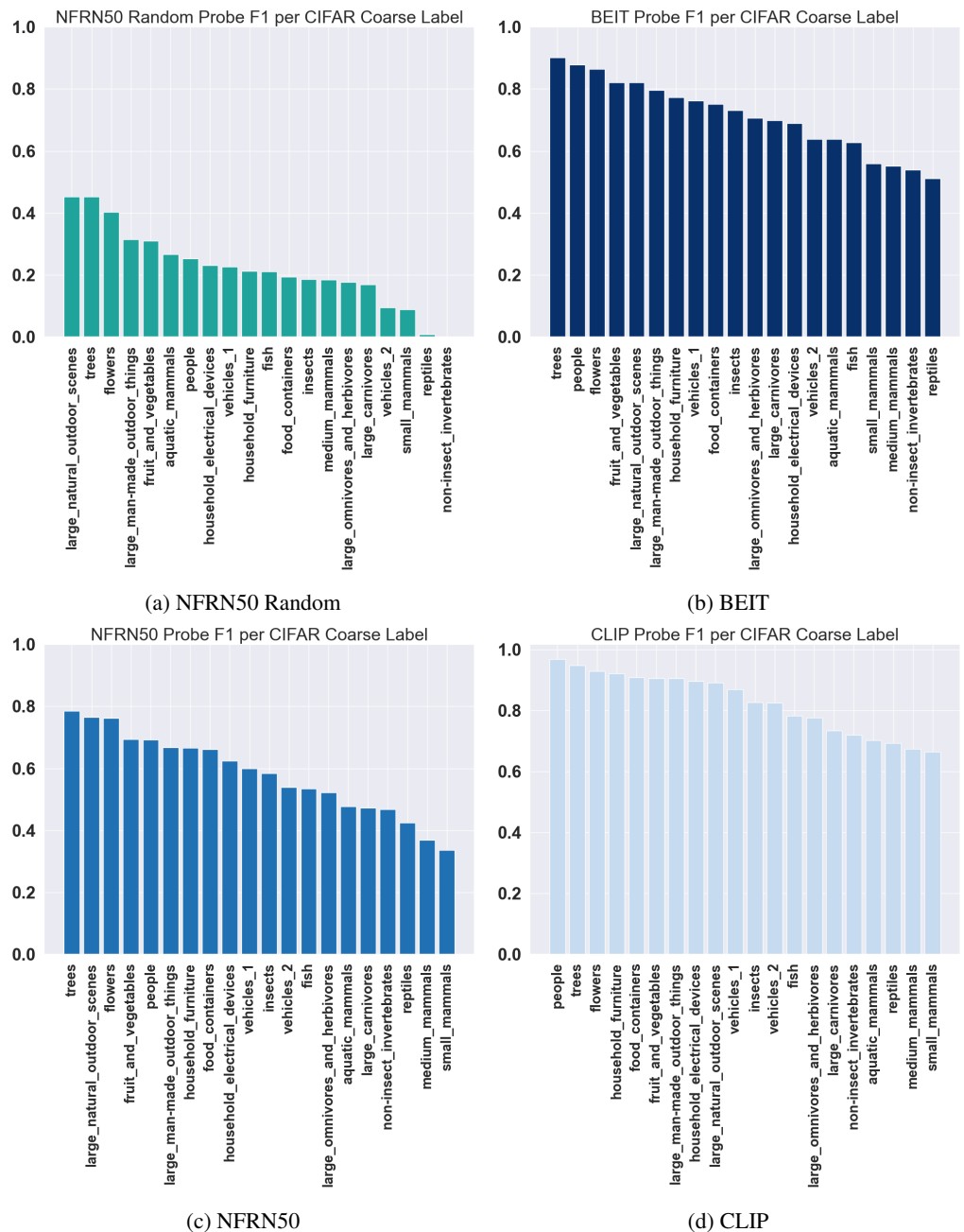

Figure 14: Probes trained on CIFAR images to classify the coarse labels of the objects in the images

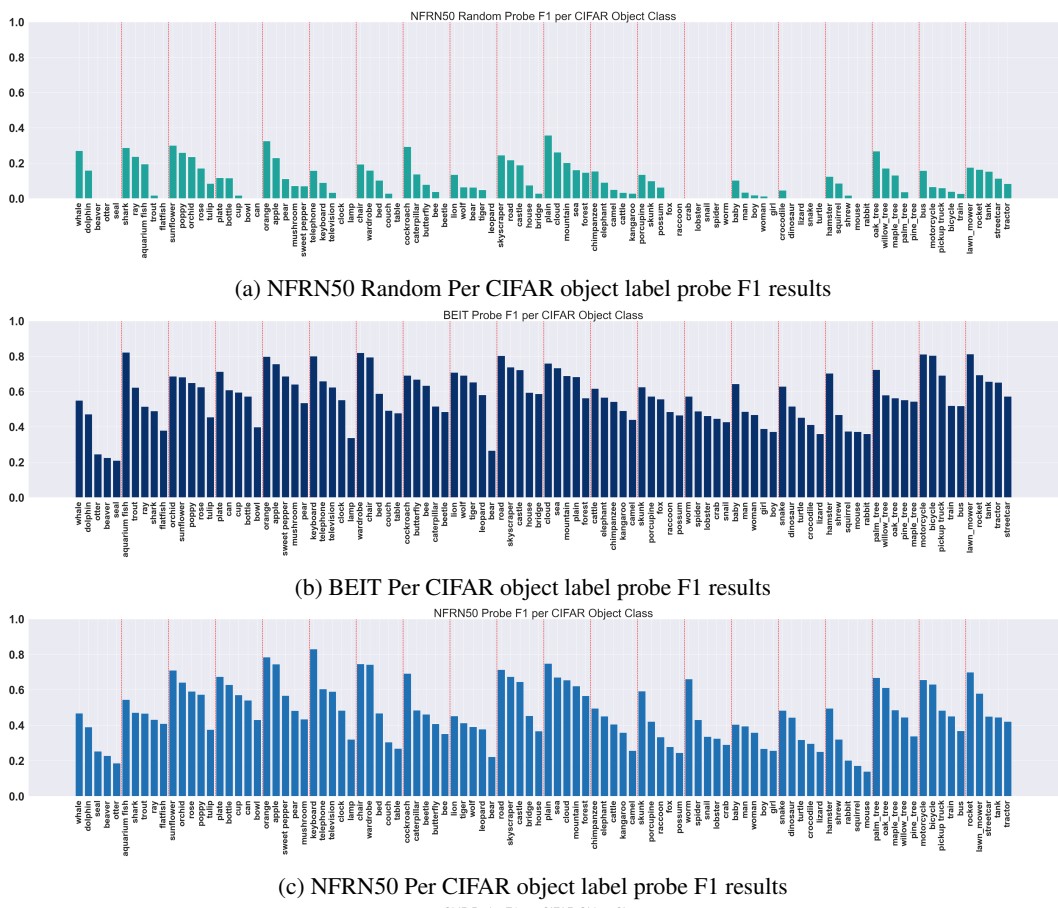

(a) NFRN50 Random Per CIFAR object label probe F1 results

(b) BEIT Per CIFAR object label probe F1 results

(c) NFRN50 Per CIFAR object label probe F1 results

(d) CLIP Per CIFAR object label probe F1 results

