# OpenReview forum: "Linearly Mapping from Image to Text Space"
_ICLR.cc/2023/Conference — ICLR 2023 poster_

### Official Review · Reviewer_N4fG · 2022-10-14

**Confidence:** 3
**Clarity, Quality, Novelty And Reproducibility:** Seems clear and novel to this reviewer.
**Correctness:** 4
**Technical Novelty And Significance:** 3
**Empirical Novelty And Significance:** 4
**Recommendation:** 8

**Strength And Weaknesses:**

Strengths:
* This paper makes a simple yet important extension to the Magma / Frozen works -- adding the simplest possible adaption layer from an image-only encoder to a language model. The simplicity also means that there are plenty of opportunities to probe what the underlying image encoder learns during pretraining. For that reason, I think this paper could be informative to researchers who work in a variety of different communities. The model also could be a great baseline for further vision-language models.
* The analysis about what fine-grained concepts are learned by the various models seems very detailed and interesting to this reviewer. The methodology could be useful for other works that want to probe these models, and to perhaps make them better.

Weaknesses:
* To this reviewer, the paper could be improved if it considered different model sizes (e.g. larger or smaller CLIP models or supervised resnets or BEITs). An interesting question is whether the probing results get better or plateau with model size (of both the encoder and perhaps the decoder; which here is GPT-J (6B parameters).

Miscellaneous: I think there is something weird with the font in this paper.

**Summary Of The Paper:**

This paper studies to what extent models trained on image-caption-only data can be frozen, passed through a linear projection, and then provided to a generative LM to generate captions for images. This builds off some earlier models/ideas like Frozen (Tsimpoukelli et al 2021) and Magma (Eichenberg et al 2021),  except here, only one linear projection is learned -- mapping an image encoder to the vocabulary space of a language model. In contrast, Frozen finetunes the whole image encoder, and Magma uses adapter layers to minimally finetune the image encoder.

The paper studies how well various image encoder learning paradigms can be used here -- BEIT (which was trained only on image patches), imagenet supervised training, and CLIP. When CLIP is plugged in here, CLIP performs well across the board on VQA and image captioning; even outperforming the authors' tuned implementation of Magma in a 0shot setting on VQA.

The paper dives further and probes how well concepts transfer (e.g. nouns, modifier, relations on COCO; perceptual concepts from Animals with Attributes).

**Summary Of The Review:**

Overall, to this reviewer, this paper seems impactful and I'd vote to accept it -- would still be curious as to how well models of different sizes do under this evaluation.

---

> ### Author Response · Authors · 2022-11-19
> **Thank you for the review and ideas regarding different model sizes**
>
> “To this reviewer, the paper could be improved if it considered different model sizes (e.g. larger or smaller CLIP models or supervised resnets or BEITs)”
> Thank you for this suggestion. We would love to do this but are unfortunately constrained by computational resources and cannot run the experiments needed to do a full analysis by model size. However, we added two new experiments with BEIT models: one that was supervised through finetuning on ImageNet22k, and one randomly initialized, as a comparison to the random NFRN50 model (see Tables 1 and 3). We think that the finetuned BEIT mostly addresses your concern, and the results give a lot of support to the idea that the linguistic supervision is playing a large factor in the performance of transfer to the LM.

---

> > ### Comment · Reviewer_N4fG · 2022-11-24
> > **Thanks for the response! I'm still positive about this paper**
> >
> > Thanks for adding the new experiments with BEIT models! This helps, but I'm also curious as to if *smaller* models would show a steep performance decrease. For instance, maybe if you don't have the compute needed to study the behavior at large scale, maybe you could extrapolate it from performance at smaller scale. If possible I think that experiment would strengthen the paper even more.
> >
> > Besides that I still think this is a strong paper so I vote to accept it. I've also read the other reviewers concerns -- I think the concern about novelty is a bit overblown and that this paper is an important extension of Frozen/Magma that the community ought to know about.

---

### Official Review · Reviewer_kxWt · 2022-10-24

**Confidence:** 4
**Correctness:** 4
**Technical Novelty And Significance:** 3
**Empirical Novelty And Significance:** 3
**Recommendation:** 8

**Clarity, Quality, Novelty And Reproducibility:**

The paper is very well written and clear, and I find the experiments to be appropriate to answer the hypothesis. Reproducibility is addressed by sharing code to reproduce experiments, and in terms of novelty the paper contributes a small data point to a series of works in frozen / soft prompts in vision and language models. Even though it is a small data point, I still find it valuable.

**Strength And Weaknesses:**

Strengths:
- Paper has a clear hypothesis and story.
- Paper's results are compelling and substantiated by well-thought experiments, which cover the important sources of variation that affect the hypothesis laid out.
- Good balance between empiricial decisions and theories to support those decisions.


Weaknesses:
- None that I can think of.

**Summary Of The Paper:**

The paper proposes a simple idea to test a clearly-defined hypothesis: whether representations learned by a vision model (trained on images only vs. trained on images including limited degrees of linguistic supervision) are functionally equivalent to those learned by a language model (up to a linear transformation). The authors report on different experiments and show that the degree of linguistic supervision used in the (frozen) image encoder matters, and the more linguistic supervision leads to better results on captioning and VQA. In turn, this is interpreted as the extent to which visual representations are functionally equivalent to textual representations (up to the linear transformation).

**Summary Of The Review:**

Comments and questions:

- Figure 4 has no number.

- Why did you not have a BEIT-Large model random initialised as a baseline? That would have added much information to the NFRN50 random baseline, since it would allow you to compare both BEIT-Large and BEIT-Large random in a similar fashion to what you did with NFRN50.

- I suggest clearly defining what is meant by conceptual/visual properties vs. lexical categories already at the introduction. You state early on in the paper that you find that all three models distinguish the former, but the more textual supervision models have during pretraining, the better models distinguish the latter. It would be clarifying to have one- or two-sentence explanations of what is meant by these properties/categories, perhaps including one or more examples of each.

- Page 9: "By looking at failure cases for each model, we can establish whether each model is predicting the presence of a similar animal or not. In Figure 4a, we show that when captions generated from each model mistake one animal for another, the mistaken animals are highly similar to the ground truth animal when measuring both
Wu-Palmer similarity (Averages: BEIT: 0.8, NFRN50: 0.81, CLIP: 0.8) and overlap of AWA properties (Averages: BEIT: 0.62, NFRN50: 0.68, CLIP: 0.59)" -> There is an implicit 'if' statement in this part of the text that troubles me. You say that "when captions mistake one animal for another" something happens. But how often do models make a mistake that is of a different nature, e.g., generate an spurious caption? Does BEIT do that more often than NFRN50 and CLIP? I would like to see these numbers.

Typos and presentation:

- "k is determined by the architecture of [the] E"

- In the Appendix, some of your Tables should be resized.

- Page 19: "All probes are [trianed]"

- The last Figure in your Appendix has no number.

- In your appendix, you include many figures. Perhaps it may be a good idea to try to be more concise?

---

> ### Author Response · Authors · 2022-11-18
> **Thank you for the review and the ideas**
>
> “Why did you not have a BEIT-Large model random initialised as a baseline?” Good question! We were constrained by computational resources when we submitted, but added it in the revision to Section 4, Table 1 (the captioning performance table). We also added an additional variant where we train a linear projection from a frozen BEIT that was finetuned on image classification (same task as NFRN50). Please see the captioning results in Tables 1 and 3 for both of these new models. We are excited about these results because they strengthen our claim that it is the linguistic supervision of the pretraining task that plays a big role in performance, not the architecture.
>
> “You say that "when captions mistake one animal for another" something happens. But how often do models make a mistake that is of a different nature, e.g., generate an spurious caption? Does BEIT do that more often than NFRN50 and CLIP? I would like to see these numbers.” -> That’s a good idea, we did find that BEIT made more mistakes than CLIP or NFRN50 in general, and the error analysis currently factors in mistakes with non-animals (they have very low Wu-Palmer scores). We rewrote the Figure 4 caption and text describing those results in the main body to make this more clear. We included a new table (Table 9 in Appendix D) which shows the per-animal and mean exact match accuracies for each model on predicting the correct animal. We think that this table, in addition to the analysis we currently have, paints a more complete picture which addresses this question.

---

### Official Review · Reviewer_s1Pu · 2022-10-30

**Confidence:** 4
**Correctness:** 4
**Technical Novelty And Significance:** 2
**Empirical Novelty And Significance:** 2
**Recommendation:** 5

**Clarity, Quality, Novelty And Reproducibility:**

The writing is good. The code should be easy to reproduce and the analysis/experiments are solid. But the novelty is limited based on the weaknesses I mentioned above.

**Strength And Weaknesses:**

Strength:
1. How to bridge pre-trained image models and pre-trained LM is quite an interesting and intriguing problem. And it can enable desirable downstream tasks such as few/zero-shot VQA utilizing stored knowledge from both worlds.
2. This paper verified a strong hypothesis: training only a linear layer is enough for mapping visual pre-trained knowledge to text space.
3. In experiments, this paper goes through recent popular visual pre-trained models and brings a relatively comprehensive analysis.

Weakness:
1. A major concern is the novelty of the paper. As admitted in this paper, Frozen also adds and tunes a linear mapping layer to process image encoder output features for LM input. The only technical difference against Frozen is that this paper doesn't update visual encoders. It's a good finding but not enough for publication at ICLR.
2. Performance vs MAGMA. In Tab1, when reporting image captioning results, only MAGMA(released) and MAGMA(ours) are compared while MAGMA(reported) is missing. If compared with MAGMA(reported), the proposed method in this paper would perform worse in NoCaps on four metrics. On the other hand, for VQA, the best result of MAGMA is from RN50x4. But in this paper, author(s) compares results on RN50x16 and the proposed method still underperforms on 1-2-4 shot VQA.
3. It's already found in other works that CLIP is better than other non-text-guided visual pre-trained models in language/multimodal understanding [1]. What this paper found echoes similar findings before.

Ref:
[1] Shen, Sheng, et al. "How Much Can CLIP Benefit Vision-and-Language Tasks?." ICLR 2022.

**Summary Of The Paper:**

The paper studies whether learning a linear layer is good enough to map image features to text space. Three different image encoders are tested and it's found CLIP performs best on tasks requiring fine-grained category visual information.

**Summary Of The Review:**

Overall, I lean toward rejection because of limited novelty and low performance.

`After Rebuttal:`

I agree that it's an interesting finding that people should know. But it doesn't surprise me because as I said, many papers have explored this direction, and this finding shares similar intuition as [Shen, et al.]. Overall, I would raise my score to weak reject but I am also okay with accepting it.

---

> ### Author Response · Authors · 2022-11-18
> **Thank you for the review and notes**
>
> “A major concern is the novelty of the paper… The only technical difference against Frozen is that this paper doesn't update visual encoders” -> Although the architecture is similar to Frozen, the novelty of our paper comes from the ability to interpret the representation spaces of these models in a new way. As soon as any model parameters are tuned, it becomes nearly impossible to make claims about the representations of the two pretrained models with performance on tasks alone. The main finding in our paper is not “we can tune fewer parameters and maintain performance”, but rather *how we can understand the relationship between the representations learned by language models and vision models*. Although we mentioned Shen, et al. in our related work, we had not expanded on the connection between our results and theirs. We changed this in the revised version to address this concern. We make it clear that our work expands on that paper: in particular, we provide evidence for a strong similarity between pretrained vision model and LM representations, and show that the linguistic supervision of the vision model pretraining objective correlates with the degree of similarity. The second part of our results section looks directly at which conceptual categories can linearly transfer between vision models and language models as a measure of how similarly the two models encode each concept. We hope our work spurs interest in understanding how models trained on different modalities represent a concept similarly/differently from one another.

---

### Official Review · Reviewer_Sxvp · 2022-10-31

**Confidence:** 4
**Correctness:** 3
**Technical Novelty And Significance:** 2
**Empirical Novelty And Significance:** Not applicable
**Recommendation:** 6

**Clarity, Quality, Novelty And Reproducibility:**

Clarity, quality and reproducibility look overall good. The novelty is rather limited.

**Strength And Weaknesses:**

Pros:
1. The idea of linear transforming is interesting.
2. The comparisons between BEIT and other (text-)supervised models are interesting.
Cons:
1. Significantly lack of discussion with relevant work. These works are highly relevant and even somewhat similar to the proposed method. Therefore, it is necessary to provide more comprehensive literature review to clarify the distinction with these work.
a. [r1,r2,r3,r4] reports a line of alternative method of leveraging pretrained (text-)supervised models for visual recognition: directly using text words to represent the information in visual input, which is a highly relevant baseline to the proposed method: they all use very few parameters or even no new parameters to bridge vision and text.
b. [r5] reports the possibility of reading out the visual information from any pretrained vision only models by tuning a text encoder with the image model fixed through contrastive learning.

2. The core assumption "structural similarity between the two spaces" is not directly tested. This could be checked by simply compare the vision similarity graph of objects from a few categories and the text similarity graph of the text embeddings of those categories.

3. The authors assumed the linearity of the transformation from vision space to textual space, which is certainly not the case in the real world as the reviewer believe simple contradictions could be found: especially when there are non-visual correlation between two words, the distance between vision space could be non-correlated with the distance in text space. For example, blue and a depressed person.
Therefore, the review takes the proposed idea as an approximation of the real world. But the author didn't provide any proper-upper bound for this.
a. For example, the linear transforming could be progressively added to a MLP.
b. Tuned MAGMA is not considered as a proper upper-bound because this is essentially a trade-off between non-linearity and the number of examples that could be used to train the non-linear mapping.
c. MAGMA also reports results on more datasets, which should be ideally also provided for the proposed method.



Minor:
Figure 5, the font size and font seem to be inconsistent.

[r1] VX2TEXT: End-to-End Learning of Video-Based Text Generation From Multimodal Inputs, CVPR 2021

[r2] Socratic Models: Composing Zero-Shot Multimodal Reasoning with Language

[r3] Visual Clues: Bridging Vision and Language Foundations for Image Paragraph Captioning

[r4] Language Models with Image Descriptors are Strong Few-Shot Video-Language Learners

[r5] LiT : Zero-Shot Transfer with Locked-image text Tuning

**Summary Of The Paper:**

In this paper, the authors proposed to learn a linear mapping from vision embedding space to text embedding space of a pretrained language model. The authors provided in-depth analysis and comparison between vision-only pretrained models and (text-)supervised vision pretrained models. The effectiveness is shown through baseline comparisons.

**Summary Of The Review:**

Overall, the reviewer thinks the analysis provided in this paper could be possibly helpful for the community but currently several cons need to be resolved towards a publication at ICLR.

---

> ### Author Response · Authors · 2022-11-18
> **Thank you for the references and good ideas**
>
> discussion of relevant work -> Thank you for these pointers to related papers, we have updated the draft to include all of these in Section 2 (Related Work). This is a very rapidly evolving area that cross-cuts language and vision communities, and so it is difficult to track down all related papers. We very much appreciate these new pointers!
>
> “The core assumption "structural similarity between the two spaces" is not directly tested. This could be checked by simply compare the vision similarity graph of objects from a few categories and the text similarity graph of the text embeddings of those categories.” -> Good idea! See Appendix B Figure 5 (and the corresponding writeup) for a brief and preliminary analysis on this using RSA to directly measure the “structural similarity” (we limit our analysis to animals to make the analysis feasible in the time). We think that this requires a more extensive followup to better understand the relationship between the two spaces. We would be interested in exapanding on this in the camera ready version. We changed some of the language surrounding these claims throughout the paper.
>
> “The authors assumed the linearity of the transformation from vision space to textual space” -> We do not assume this. This was a hypothesis we tested (which was based on some initial insights we’d seen while exploring other approaches to cross-modal transfer).  We agree with you that it is not intuitive and were ourselves surprised that our results turned out as positive as they did. We think that’s what makes the finding interesting! With regards to a better “upper-bound”, we should emphasize that the goal of this work was _not_ to outperform existing approaches. Rather, it was to test (using standard evaluation metrics) whether image encodings could be viewed as “the same”  as text encodings up to a linear transformation, and to compare image encoders with differing amounts of linguistic supervision. MAGMA is included as a point of reference to show how much is lost by constraining the transformation to be linear. (As it turns out, not much.) Since MAGMA is an effective image captioning model, we believe that the comparison is valid. But we don’t claim it’s an “upper bound”. Surely new models will be introduced which perform better than MAGMA and thus much better than our linear projection.
>
> Thank you for a very thorough review, for including missing references, and for actionable followup experiments to make our work more complete. We hope that our revisions make our results more clear and compelling.

---

### Decision · Program_Chairs · 2023-01-20

**Decision:**

Accept: poster

**Justification For Why Not Higher Score:**

This paper, while excellent in its scientific approach, does not propose something fundamentally new. In addition, given the score, a poster presentation would be appropriate.
(Having said that, spotlight wouldn't be that bad as the message of the paper is catchy.)


**Justification For Why Not Lower Score:**

This paper received split scores of 3,6,8,8. All reviewers expressed interest in the linear mapping approach and its excellent performance. On the other hand, concerns were raised about the lack of novelty of the method, lack of comparison with other studies, and the obviousness of the results. To address these issues, the authors added explanations and some new results in the response, but the focus was still on novelty.
In the AC’s opinion, the novelty and contribution of this paper do not lie in the technique itself of learning linear mapping and its good performance, but in the approach of bridging independently learned vision encoders and language decoders with mapping only, and the new benefits it brings to the study of representation learning are well demonstrated by experiments as highlighted by two reviewers. Linear projection is its simplest form and would be an excellent baseline in this approach.
On the other hand, in relation to this point, the AC thinks the current paper is somewhat misleadingly written, as if it asserts the linear relationship between vision and language spaces. The paper should clearly convey the motivation to find out how far linear projection as the simplest form of mapping can go (and the results have been very good empirically).
Overall, the AC believes that the above problems are relatively easy to correct and that this paper will bring new insights and methodologies that researchers in the fields of vision-and-language and representation learning ought to know. In this sense, the pros of the paper outweigh the cons well, thus the AC would like to recommend acceptance.


**Metareview: Summary, Strengths And Weaknesses:**

Summary:
In this paper, the authors propose an approach to learn a mapping projection from a pretrained vision encoder to a text decoder (pretrained language model) while the parameters of those encoders and decoders are frozen. Although the technique itself might seem trivial from an engineering viewpoint, this paper deeply addresses a clearly-defined scientific hypothesis: whether representations learned by a vision model are functionally equivalent to those independently learned by a language model (up to a linear transformation). The paper presents a somewhat surprising results that just learning a single linear layer is as good as the previous methods based on fine-tuning or non-linear adapters in terms of the aligning visual features to text space. The authors demonstrated that this simplicity (mapping approach) also enables that there are plenty of opportunities to probe what the underlying image encoder learns during pretraining and how they can be transferred. Where, the authors conducted systematic experiments and show that the degree of linguistic supervision used in the (frozen) image encoder matters, and the more linguistic supervision leads to better results on captioning and VQA.

Strengths:
1. This paper investigates an intriguing approach that learns a simple mapping to bridge pre-trained image models and pre-trained LM.
2.  This approach enables to deeply analyze what kind of features are learned by an individually-trained visual encoders and how they can be transferred, as demonstrated in the paper.
3.  The surprisingly good result with a linear projection is thought-provoking, and can be a firm baseline in vision-and-language studies.
4.  The experiments are well-organized and comprehensive. The authors tested vision encoders with increasing levels of linguistic supervision in pretraining, which is reasonable and interesting.
5. In particular, the methodology of evaluating what kind of (fine-grained) concepts are learned and transferred by the various models seems useful for other works that want to probe pretrained models.
6. Reproducibility is addressed by sharing code to reproduce experiments.

Weaknesses:
1. From a technical viewpoint, the novelty of the method is little since it can be viewed as a simplified version of the standard methods like fine-tuning or adapters, although the simplicity does not directly mean bad.
2. Direct evaluation regarding the structural similarity between the vision and language spaces can be enhanced more.
3. The writing of the paper is a little bit misleading in its claims.





**Note From Pc:**

if the above contains the word "oral" or "spotlight" please see: "oral" presentation means -> notable-top-5% and "spotlight" means -> notable-top-25%. As stated in our emails, we are disassociating presentation type from AC recommendations